# Vertically stacked monolithic perovskite colour photodetectors

Sergey Tsarev[1,2], Daria Proniakova[1], Xuqi Liu[1], Erfu Wu[1,3], Gebhard J. Matt[1,2], Kostiantyn Sakhatskyi[1,2], Lorenzo L. A. Ferraresi[1,3], Radha Kothandaraman[2], Fan Fu[2], Ivan Shorubalko[3], Sergii Yakunin[1,2✉] & Maksym V. Kovalenko[1,2✉]

Modern colour image sensors face challenges in further improving sensitivity and image quality because of inherent limitations in light utilization efficiency[1]. A major factor contributing to these limitations is the use of passive optical filters, which absorb and dissipate a substantial amount of light, thereby reducing the efficiency of light capture[2]. On the contrary, active optical filtering in Foveon-type vertically stacked architectures still struggles to deliver optimal performance owing to their lack of colour selectivity, making them inefficient for precise colour imaging[3]. Here we introduce an innovative architecture for colour sensor arrays that uses multilayer monolithically stacked lead halide perovskite thin-film photodetectors. Perovskite bandgap tunability[4] is utilized to selectively absorb the visible light spectrum's red, green and blue regions, eliminating the need for colour filters. External quantum efficiencies of 50%, 47% and 53% are demonstrated for the red, green and blue channels, respectively, as well as a colour accuracy of 3.8% in $\Delta E_{\text{Lab}}$ outperforming the state-of-the-art colour-filter array and Foveon-type photosensors. The image sensor design improves light utilization in colour sensors and paves the way for the next generation of highly sensitive, artefact-free images with enhanced colour fidelity.

Obtaining an image that accurately reproduces human colour perception poses a significant challenge, primarily because the photodetector elements in digital camera sensors do not interpret colours in a manner analogous to human vision[5]. To achieve colour imaging, an array of band-pass optical filters, designated as a colour-filter array (CFA), is positioned atop a monochrome sensor to separate the visible light spectrum into its red, green and blue (RGB) components[2] (Fig. 1a,b). After the collection of raw input data, the colour image is reconstructed using interpolation algorithms[6]. This technique approximates human colour perception but it still has multiple drawbacks[7–9]. The drawbacks include demosaicing artefacts and large optical losses as the CFA must absorb approximately two-thirds of the light spectrum incident on each pixel (Fig. 1b). As a result, the external quantum efficiency (EQE) is reduced when averaged over a full-colour set of 4 pixels (that is, a photosite; ref. 10 and Fig. 1a,c).

To address the inherent limitations of the CFA, Sigma Corporation has developed an alternative image sensor architecture, known as Foveon. The Foveon-type photosite, shown in Fig. 1d, has a distinctive structure with three integrated vertically stacked pixels that collect RGB data without filtering losses inherent to the CFA[11,12]. Although Foveon-type sensors are more efficient in light utilization and thus have a higher EQE (Fig. 1f), they face challenges in accurate colour perception[3], which are only partially addressed with elaborate post-processing algorithms. These difficulties arise from the silicon active layer's nearly uniform sensitivity across the visible spectrum, providing no inherent wavelength selectivity. Its spectral selectivity is afforded by silicon's steeply wavelength-dependent absorption coefficient and, hence, penetration depth[3], as illustrated in Fig. 1e. To achieve optimal colour perception, an ideal Foveon-like sensor should have active layers specifically responsive to the red, green and blue regions of the visible spectrum. This necessitates the use of non-silicon materials designed for targeted absorption[13–16].

Lead halide perovskites APbX$_3$ ('A' being formamidinium (FA), methylammonium (MA) or caesium cations, and 'X' being Cl, Br or I) present a compelling class of semiconductors for multilayer sensors, owing to their optical and electronic characteristics affording highly sensitive thin-film photodetectors for the visible range[13,17–20]. Facile mixing of bromides with iodides or chlorides into the respective solid solutions[21] allows adjustments of their bandgap energies in the 1.6–3 eV range[4]. Integrated perovskite layers with carefully optimized bandgaps may thus serve as sequential optical filters capable of selectively absorbing blue, green and red wavelengths (Fig. 1g,h).

Such perovskite multilayer sensors have thus far been discussed theoretically[22,23], but the practical implementations were limited to only the mechanical stacking of three individual detectors[13,24]. Monolithic, thin-film multilayer stacks are, however, unavoidable for future image sensor arrays, owing to the lithographic patterning and integration into readout circuitry. In this study, this critical milestone of multilayer fabrication (Fig. 1h) is demonstrated through the focus on all-evaporated perovskite layers and judicious choice

[1]Laboratory of Inorganic Chemistry, Department of Chemistry and Applied Biosciences, ETH Zürich, Zurich, Switzerland. [2]Laboratory for Thin Films and Photovoltaics, Empa – Swiss Federal Laboratories for Materials Science and Technology, Dubendorf, Switzerland. [3]Transport at Nanoscale Interfaces Laboratory, Empa – Swiss Federal Laboratories for Materials Science and Technology, Dubendorf, Switzerland. ✉e-mail: yakunins@ethz.ch; mvkovalenko@ethz.ch

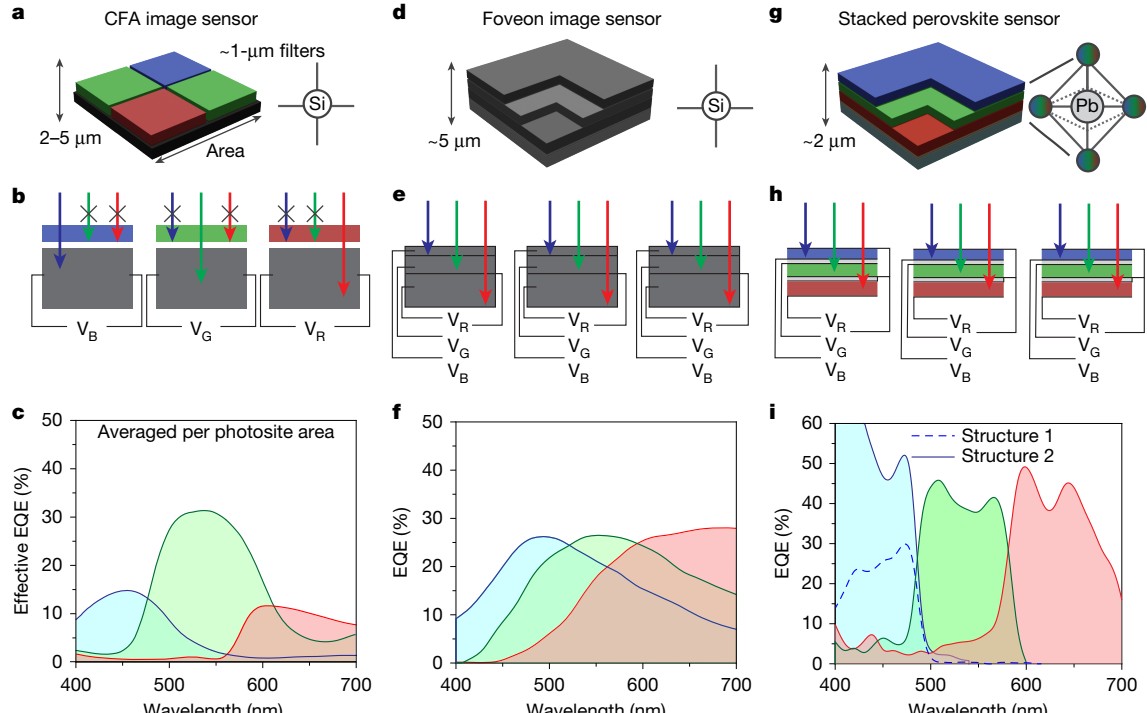

**Fig. 1 | Schematics, architecture and EQE spectra of different image sensor architectures. a–c**, A CFA image sensor. Each photoactive site (photosite) comprises 4 pixels (**a**), selectively sensitive to the green (G), blue (B) and red (R) spectral regions of visible light. Light absorbed by the optical filters does not contribute to the photogenerated signals $V_B$, $V_G$, $V_R$ (**b**), leading to a reduction in the EQE when averaged across the entire photosite (**c**). Data were recalculated from the Sony IMX249 datasheet[10] by dividing the blue pixel and red pixel EQE by 4 and the green pixel EQE by 2. **d–f**, A Foveon sensor. Each photosite consists of vertically stacked silicon photodetectors (**d**). In the absence of colour filters, colour selectivity is achieved through silicon layers of different thickness, leveraging the strongly wavelength-dependent penetration depth of light in this indirect-bandgap material (**e**). This architecture results in a substantially higher quantum efficiency per photosite compared with the CFA sensor, albeit at the cost of reduced colour selectivity (**f**). Data from ref. 3. **g–i**, A stacked-perovskite sensor, developed in this work. The perovskite stacked detectors are monolithically integrated (**g**), serving as active optical filters for the underlying stacks (**h**). This design achieved substantially higher quantum efficiency and colour accuracy than Foveon-type and CFA sensors (**i**). Data were experimentally obtained from structure 1 (B, dashed line) and structure 2 (R, G and B); see details of tested detector structures in Supplementary Note 1.

of transport and dielectric materials. The obtained devices show a nearly two-times-higher averaged EQE compared with conventional CFA technology (Fig. 1i). Beyond their readily tunable bandgaps, perovskites may potentially act as a more technologically simple alternative to current state-of-the-art back-side illuminated complementary metal–oxide–semiconductor (CMOS) image sensors for maximization of active sensor area[25,26]. The high extinction coefficient of lead halide perovskites allows complete absorption of the visible light spectrum by just a few-hundred-nanometres-thick film[27], favouring the use of wide-angle lenses and their closer positioning of lenses to the sensor, and enhanced sensitivity, particularly in the red region of the visible spectrum (compared with Foveon technology)[28]. The stacked sensor technology, where the distance between the layers can be additionally controlled by dielectric spacers, also allows for the correction of chromatic aberration (that is, when the focal plane for blue colour is a few to a few tens of micrometres closer to the objective than that for red colour). Furthermore, our sensor architecture eliminates demosaicing artefacts and enhances effective spatial resolution owing to the vertical stacking of pixels.

## Stacked-detector structure

The fabricated sensor architecture is illustrated in Fig. 2a, depicted as a cross-section image of the device with annotated layers corresponding to each colour stack designed for the sequential absorption of blue, green and red regions of the visible light spectrum. Each colour stack is constructed as an individual heterojunction photodiode using composite charge-transport layers, transparent electrodes and active perovskite absorber layers. To circumvent the dissolution of the previously deposited perovskite absorber layers during the deposition of the subsequent perovskite layers[29,30], all perovskite materials were co-evaporated in a specialized physical vapour deposition chamber, yielding homogeneous pinhole-free films[31] (Supplementary Fig. 1). To achieve colour-specific absorption in different-bandgap layers, we used $MAPbBrI_2$, $CsPbBr_2I$ and $CsPbBr_2Cl$ nominal active layer compositions for red, green and blue colour detection, respectively. The optical absorption and energy-dispersive X-ray spectra of the individual perovskite films are presented in Fig. 2b and Supplementary Table 3 accordingly. To prevent sputtering damage during the deposition of the transparent indium tin oxide (ITO) electrodes[32,33], we utilized protective zinc oxide or aluminium zinc oxide coatings. The pixels are separated by $SiO_x$ dielectric layers deposited through magnetron sputtering. In structure 2, we removed the sputtered dielectrics and introduced gradient growth of $CsPbBr_2Cl$ perovskite, to simplify the fabrication procedures and increase the quantum efficiency (Supplementary Note 1 and Supplementary Tables 1 and 2). The EQE of a champion device with structure 1 is shown in Supplementary Fig. 2. The full photodetector stack (Fig. 2a) is illuminated from the top, and its overall photoconversion efficiency is a complex interplay[22] of light interference, transmittance and charge-transport effects.

The resulting current–voltage (*J–V*) traces are shown in Fig. 2c–e (for structure 1) and Extended Data Fig. 1 (for structure 2 and single-colour detectors). These characteristics show photodiode behaviour with

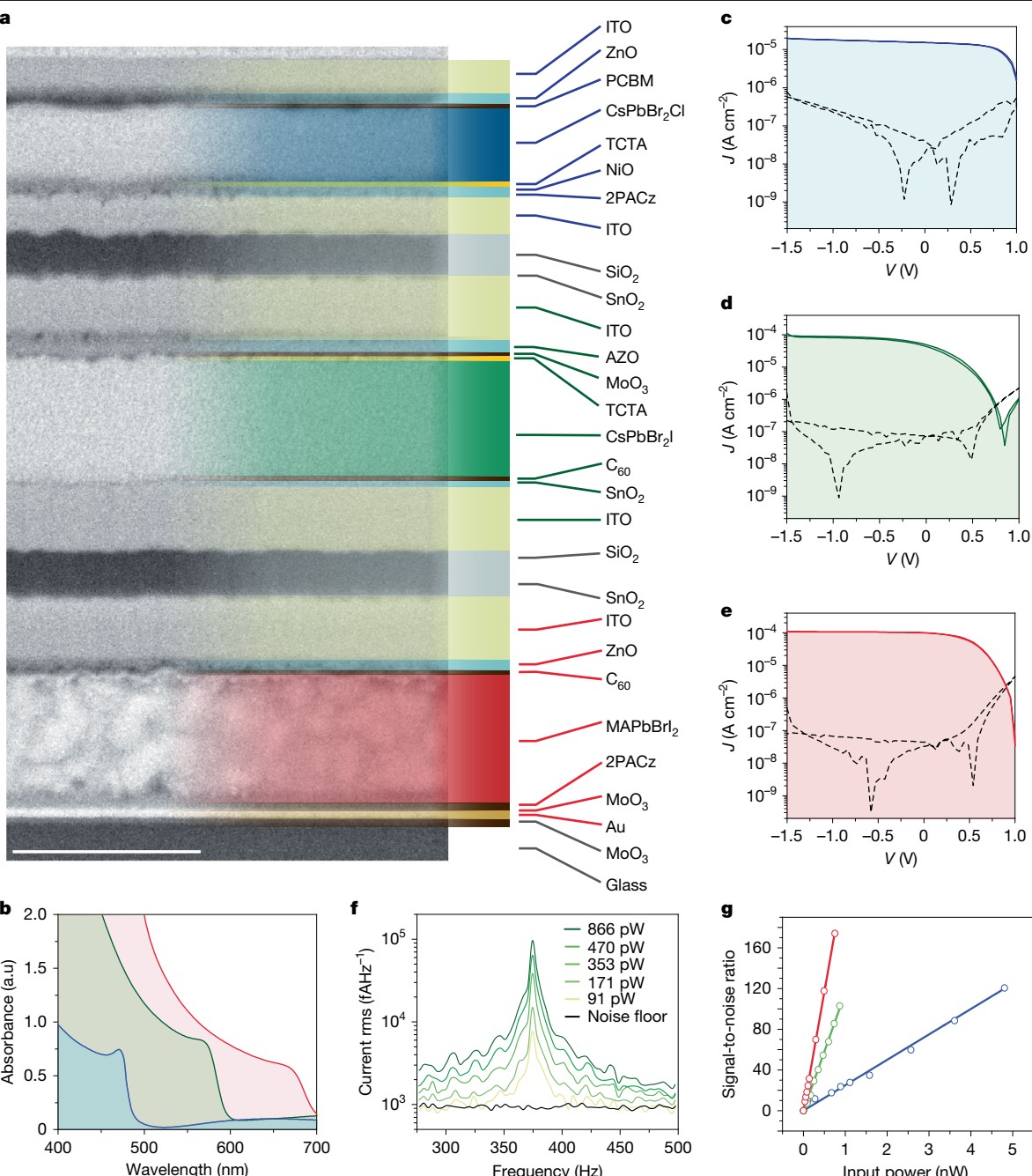

**Fig. 2 | Stacked-detector architecture and performance. a**, Cross-section scanning electron microscopy image of full-colour stacked-perovskite photodetector (photosite), structure 1. The right part of the image is colour-highlighted and labelled with the layer descriptions. Scale bar, 500 nm. 2PACz, (2-(9H-carbazol-9-yl)ethyl)phosphonic acid; PCBM, (6,6)-phenyl $C_{61}$ butyric acid methyl ester; TCTA, 4,4′,4-tris(carbazol-9-yl)triphenylamine; AZO, aluminium zinc oxide. **b**, Absorption spectra of $MAPbBrI_2$, $CsPbBr_2I$ and $CsPbBr_2Cl$ individual films used as absorber materials for R, G and B channels (pixels), respectively. **c–e**, $J–V$ characteristics (dashed lines are dark; solid lines are illuminated traces) of champion B (**c**), G (**d**) and R (**e**) colour pixels of the full-colour detector. Illumination power density was set to be 1 mW $cm^{-2}$ for each colour. **f**, The noise spectral density (in rms, root mean square) of the green channel of the full-colour detector was measured under illumination and in dark conditions. Other colour pixels are shown in Extended Data Fig. 2. **g**, Linearity of the full-colour stacked-detector response for R, G and B pixels.

good rectification, sensitivity and low dark-current densities, ranging between 10 nA $cm^{-2}$ and 50 nA $cm^{-2}$ at a −1 V reverse bias. In addition, structure 1 showed no breakdown until a −1.5-V bias, allowing zero- and reverse-bias regimes. Structure 2 showed higher EQE values compared with structure 1 at the expense of higher leakage under reverse bias for green detectors, probably owing to the removal of the dielectrics between green and blue pixels. The detectors showed a 3-dB bandwidth drop at 5.1-MHz, 12.7-MHz and 10.3-MHz detectors with 20 × 200 μm

area (Supplementary Fig. 3) and capacitance of 125 nF $cm^{-2}$, 42 nF $cm^{-2}$ and 58 nF $cm^{-2}$ at 100 Hz for red, green and blue detectors in structure 1, respectively (Supplementary Fig. 4). The noise-equivalent power of 1.6 pW $Hz^{-1/2}$, 3.2 pW $Hz^{-1/2}$ and 11.5 pW $Hz^{-1/2}$ (Fig. 2f) and corresponding specific detectivity of 1.5 × $10^{11}$ Jones, 8.6 × $10^{10}$ Jones and 2.0 × $10^{10}$ Jones were calculated for red, green and blue pixels in structure 1, respectively. Furthermore, the stacked-detector photosite showed commendable linearity for all three colours (Fig. 2g).

The detectivity observed in our data for the complex stacked architecture lags only slightly behind the best values obtained for solution-processed single-diode perovskite detectors (Supplementary Table 4). This insight underscores the potential of stacked-detector technology, highlighting a clear path towards achieving lower noise levels and possibly surpassing current benchmarks in detector performance.

The presented stacked-detector structure, although not unique to perovskites as absorbers, offers distinct benefits over other major bandgap-tunable semiconductors, such as III–V compounds (for example, InP, InAs and GaN) and organic materials. For instance, integration of III–V absorbers would necessitate high-cost epitaxial growth by molecular-beam epitaxy or chemical vapour deposition and is much restricted in the scope of substrate, typically with the epitaxial relationship[34,35]. However, organic absorbers, being similar to perovskites in terms of solution processability and facile film formation on diverse materials, typically suffer from slower response speeds[36] and lower charge-carrier mobility[37], limiting their effectiveness in dynamic imaging applications. For comparison, perovskite films are a unique class of materials featuring comparable electronic characteristics of III–V compounds and facile and versatile deposition of organic materials. Further comparison of diverse materials choices for integrated multilayer sensors can be found in Supplementary Note 2.

## Colour accuracy of stacked detectors

Colour accuracy and fidelity are the crucial characteristics of modern image sensors. Unlike broadly accepted standards for evaluating colour accuracy for colours produced by emission (for example, displays) and subtraction (printed images), there exists far more ambiguity in evaluating the colour accuracy of image sensors. Depending on whether the target application is for human or machine vision, the evaluation criteria may be vastly different. The latter case of machine vision is rather straightforward as it leverages a pragmatic principle that spectral information should be correctly assigned to distinctly separate colour channels. The criteria of colour accuracy would then correspond to the optimal spectral distribution of the channels and their minimal overlap[1,38]. Thus for the machine-vision application, the architecture of perovskite stacked photodetectors appears to be close to ideal owing to tunable additive responsivity spectra (Fig. 1i), much like the prism camera concept[39].

However, the assessment of colour accuracy in relation to human vision is far more ambiguous. Colour accuracy is often evaluated visually in a qualitative and rather subjective manner, even in professional areas such as photography or publishing, and it then also depends on the performance of the final image reproduction system (either display, projector or electronic printing). Each sensor model requires an individual sophisticated procedure to transform a matrix of device-dependent signals (that is, raw-RGB channels) into device-independent, commonly used colour spaces (XYZ, standard RGB or sRGB, Lab and so on). In this case, a transformation matrix (Supplementary Note 3) that applies device-specific interchannel subtractions is necessary to further improve the colour fidelity. Any deviation of this matrix from the diagonal one will increase the sensor noise, as noise from the other two channels will be merged into the response of the main channel.

To showcase the enhanced colour perception of the stacked photodetector, we measured its response to light reflected from the patches of a Macbeth ColorChecker chart (see Supplementary Note 3 for detailed procedures). This chart includes a set of pigmented areas covering a wide gamut of subtractive colours and is a standard tool in digital photography for the calibration of colour reproduction, illumination and white balance. Figure 3a illustrates the simplified measurement scheme used to evaluate the detector's colour fidelity. In brief, a D50 light-emitting diode (LED) light source (spectrum shown in Fig. 3b) sequentially illuminates each patch of the ColorChecker chart.

The patches reflect specific spectral characteristics (Fig. 3c) according to their pigment composition, and this reflected light is projected onto the perovskite stacked photosite. The resulting photocurrents from each RGB pixel (coloured bars, right $y$ axis in Fig. 3d) can be understood as the spectrally integrated product of the reflected light (solid line, left $y$ axis in Fig. 3d) and the EQE spectra of the corresponding colour pixels (dashed lines, left $y$ axis in Fig. 3d). In Supplementary Note 3, we use this approach to directly compare the colour accuracy of perovskite stacked photodetectors with mainstream sensor technologies using their EQE characteristics. Whether obtained by direct measurement or by EQE-based recalculation, the device-dependent RGB signals for all patches form a $3 \times 4 \times 6$ matrix (Fig. 3e). These raw signals then undergo matrix transformation to sRGB, gamma correction, white (or grey) balancing, and colour space conversion (detailed in Supplementary Note 3 and Supplementary Scheme 1 with resulting data visualized at Extended Data Figs. 3 and 4). The final data are rendered as a colour palette (Fig. 3f) for visual comparison with the original ColorChecker chart (Fig. 3a). To objectively and quantitatively compare colour imaging sensors of different technologies, we calculated the colour accuracy $\Delta E_{Lab}$ (CIE1976 standard; Supplementary Note 3, Supplementary equation (4)) using the sensor's EQE (Fig. 1c,f,i) for stacked-perovskite, CFA and Foveon-type imagers. The simplest and most evident case of unity transformation matrix $M_1$ already shows a clear advantage of the stacked sensor that delivers a $\Delta E_{Lab}$ of 11.2 (Extended Data Fig. 4b and Supplementary Table 6), whereas other types show notably larger colour errors. A few iterations of matrix $M_{corr}$ optimization allowed marked reduction of $\Delta E_{Lab}$ to 3.8 (Extended Data Fig. 4e), which remains reasonably low for various illumination conditions (Supplementary Table 7 and Supplementary Fig. 5). The experiment with measured photocurrents demonstrates $\Delta E_{Lab} = 7.6$, worse than the EQE-estimated value, which can be explained by nonlinearities and some geometrical imperfections in the stacked diodes. Notably, the low $\Delta E_{Lab}$ for the perovskite stacked sensor was obtained with a near-to-diagonal transformation matrix. Its diagonality is evaluated with so-called Frobenius norms (Supplementary Note 3 and Supplementary equation (3)) to be about 0.18. The closer the matrix is to diagonal (that is, as lower the Frobenius norms), the less 'aggressive' cross-channel correction is required and thus higher signal-to-noise characteristics will be achieved. That is particularly important for low-light intensity conditions, where Foveon ($\Delta E_{Lab} = 12.8$ for $F(M_{corr}) = 0.56$) is known to be suffering from noise issues.

## Imaging with stacked detectors

Demosaicing is a critical digital-image-processing technique used by conventional colour digital cameras to reconstruct a full-colour image from the data captured by an image sensor overlaid with a CFA. Given that each pixel receives only a fraction of the colour information, demosaicing algorithms interpolate the missing colours for each pixel by analysing the colours captured by adjacent pixels. Although this method enables colour digital imaging with sensors that do not inherently capture full colour at every pixel, it introduces several challenges, including potential resolution loss, the creation of artefacts such as colour moiré and false colours, and the need for interpolation algorithms to estimate the missing colours[40,41]. These issues can affect image quality, leading to less sharp, artefact-prone images, and necessitating additional processing power and time. On the contrary, stacked-perovskite detectors and other stacked-type sensors do not need demosaicing as they capture complete colour information at every pixel location.

To validate the expected advantages of a perovskite stacked imager for the elimination of demosaicing colour artefacts, as well as for achieving higher spatial resolution, we fabricated and characterized stacked photodetector $8 \times 8 \times 3$ pixel arrays and mechanically stacked thin-film transistor (TFT) $64 \times 64 \times 3$ detector arrays. The simplified cross-bar

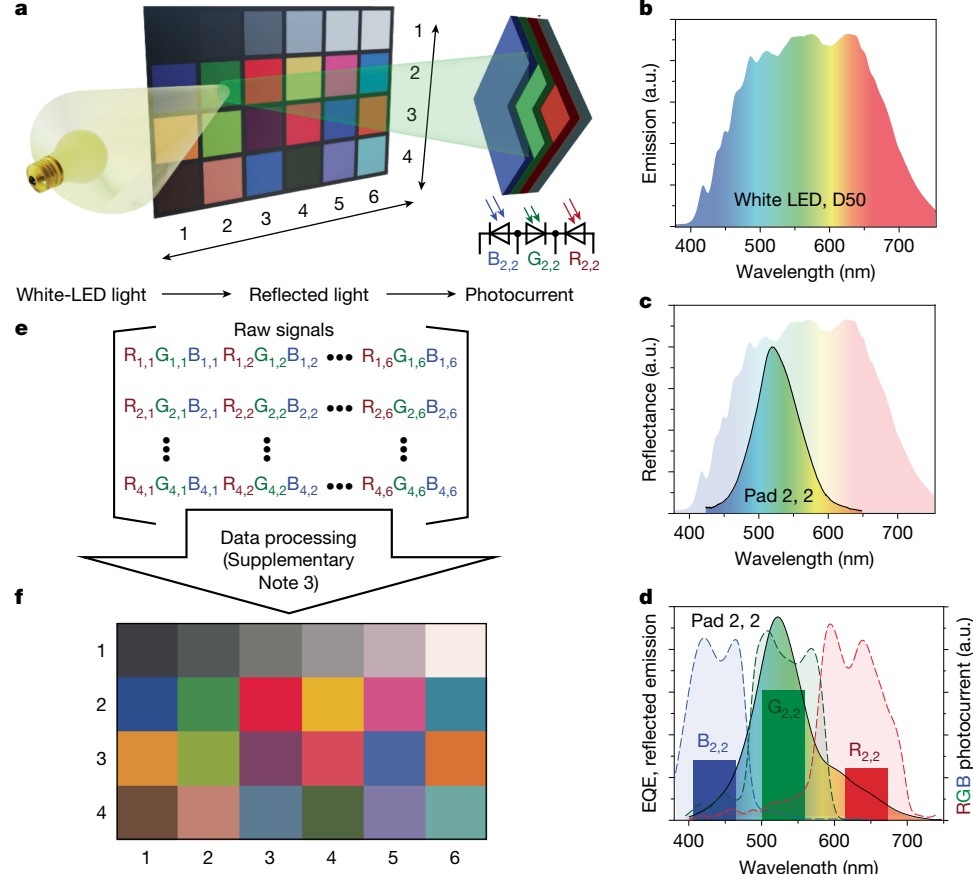

**Fig. 3 | Stacked-detector colour accuracy. a**, A colour-accuracy measurement scheme. **b**, Emission spectrum of the white LED used to illuminate the ColorChecker patches. **c**, The reflectance spectrum of a green patch [2, 2] of the ColorChecker on a background of white-LED emission. **d**, RGB photocurrent signals (coloured bars, right y axis) as spectrally integrated products of reflected light of the patch [2, 2] (solid line) and corresponding EQE spectra (dashed lines, left y axis) of the stacked detector. **e**, Data matrix of measured photocurrents (that is, device-dependent RGB signals) for each pad of the ColorChecker chart. **f**, An image of the ColorChecker chart reconstructed from photocurrents of all three colour pixels of the stacked-perovskite detector. The intermediate images representing each calculation step are shown in the Extended Data Fig. 3. a.u., arbitrary units.

layouts of electrodes for each single-colour layer of pixels (R, G or B) are shown separately in Fig. 4a, and photos of the cross-bar and TFT arrays are shown in Supplementary Fig. 7. To evaluate the impact of lateral pixelation on colour selectivity and linearity and gather corresponding statistics, we measured the photocurrents of all individual RGB pixels of the cross-bar arrays under varied red-, green- and blue-LED-light illumination (Supplementary Fig. 8). Figure 4b–d shows histograms of the photocurrents for each RGB channel to red-, green- or blue-LED light at identical photon flux normalized against the total response of the three channels. We observed that these distributions matched previously measured spectral responsivity spectra, indicating that the pixelated arrays generally retain the high degree of colour selectivity, inherent to a single stacked-type photosite.

In CIELAB (colour space defined by the International Commission on Illumination), colours are defined by three components: $L$ (lightness), $a$ (ranging from green to red) and $b$ (ranging from blue to yellow). Unlike simple RGB values, CIELAB is designed so that the numerical distance between two points closely corresponds to the physiologically perceived difference in colour by the statistically averaged human eye. This property makes it particularly useful for assessing colour accuracy because even subtle deviations in colour become quantitatively meaningful. To investigate the colour accuracy of the cross-bar array, we measured the pixel-wise distribution of CIELAB Lab colour values for the reflected light from black (1-1), white (1-6), blue (2-1), green (2-2) and red (2-3) patches of the Macbeth chart (Fig. 4e). For the blue patch, the measured CIELAB values showed a mean of $L = 4 \pm 1$, $a = 15 \pm 3$ and

$b = -35 \pm 3$. The green patch showed a mean of $L = 29 \pm 10$, $a = -30 \pm 9$ and $b = 29 \pm 9$, whereas the red patch presented mean values of $L = 19 \pm 3$, $a = 40 \pm 4$ and $b = 30 \pm 4$. In addition, the values are comparable to the values calculated from the responsivity spectra of CFA, perovskite and Foveon pixels. The cross-bar arrays statistically showed lower colour accuracy than the single-pixel detectors, which can be primarily attributed to optical and electrical crosstalk, variations and misalignment in the ITO electrode stripes. However, as we consider the cross-bar array as proof of concept of a monolithically stacked imager, the next step in the development of these sensors would be an integration with active pixel readouts. This would require substantial advancements in perovskite lithography and vertical interconnects for multilayered structures, as well as the design of dedicated triple-channel image sensor CMOS or TFT circuits. To provide a more comprehensive understanding of real-world photography using vertically stacked-detector arrays, we captured photographs of an entire ColorChecker chart using mechanically stacked RGB perovskite TFT imagers (Extended Data Fig. 5a–i, and Supplementary Fig. 9 for data processing procedure). Similar to the images reconstructed from the monolithically integrated stacked detectors, the photographs showed very good colour perception.

We utilized experimentally obtained photoresponse data from the $8 \times 8 \times 3$ perovskite sensor array or from TFT arrays to simulate Foveon or CFA operation. The stacked-type sensor was realized by using all 194 individual colour pixels obtained from the entire array. For the CFA operation, each site of the arrays used only one RGB layer with

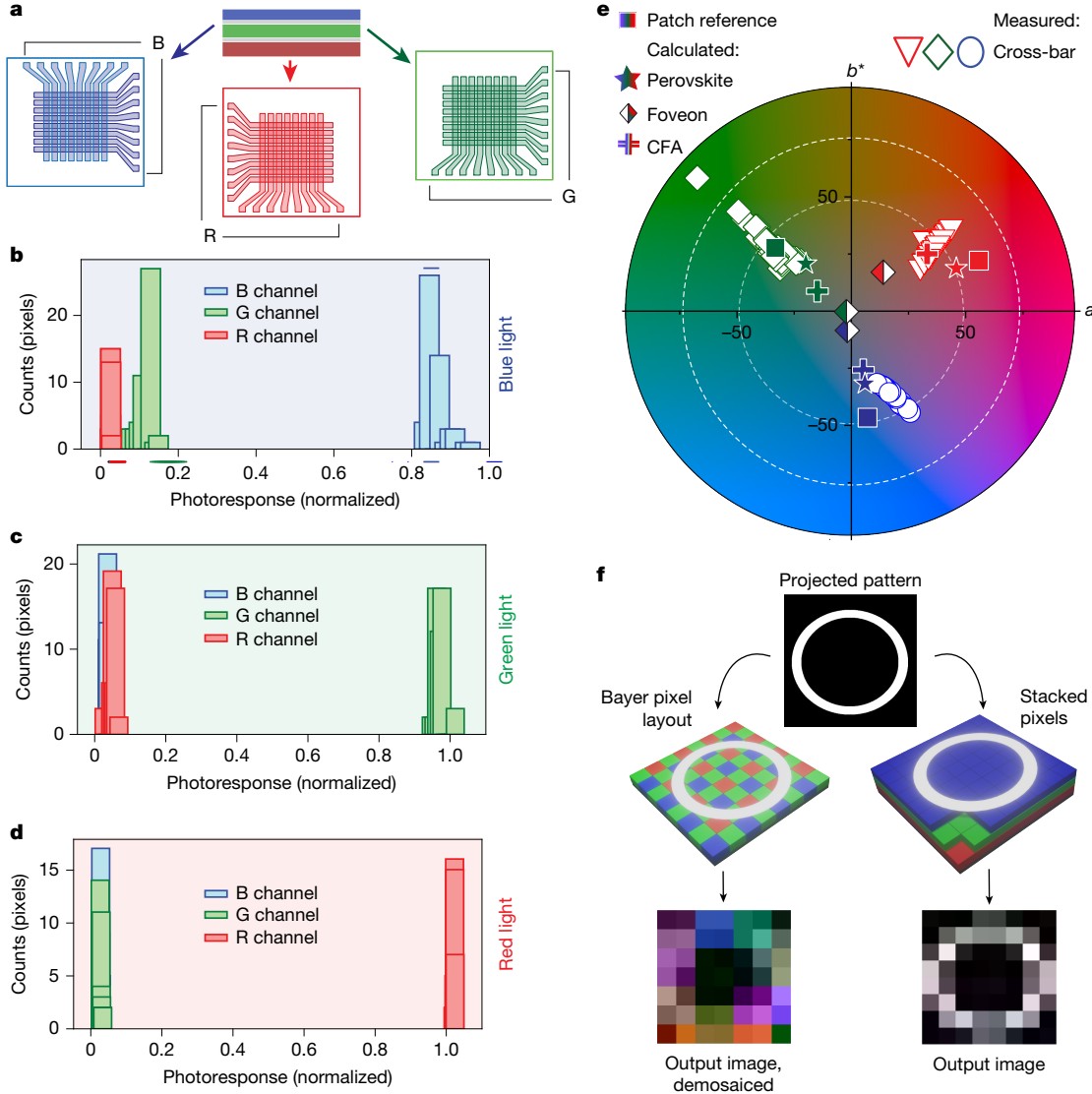

**Fig. 4 | Imaging with monolithically stacked-detector arrays in cross-bar configuration. a**, Cross-bar layouts of electrode layers for each imager array stack. The entire array is fabricated as a monolithic structure analogous to Fig. 2a. **b**–**d**, Histograms of RGB channel photocurrents in response to blue-LED (**b**), green-LED (**c**) or red-LED (**d**) light (Supplementary Fig. 6), normalized against the total response of the three channels. **e**, Distribution of cross-bar array responses to D50 illuminant light reflected from blue, green and red colour patches of the ColorChecker. The colour distribution is represented within the $a$–$b$ coordinate system, with the luminance parameter ($L$) maintained at 46%. Reference $a$–$b$ values of the colour patches are shown as squares. Theoretically calculated values for perovskite, Foveon and CFA sensors are shown as filled stars, rhomboids and crosses, respectively. **f**, An experiment with imaging a black-and-white ring pattern (top) shows the appearance of colour artefacts after the use of demosaicing algorithms for a Bayer CFA imager (left) and the absence of colour artefacts when using a stacked-perovskite imager (right). **b**–**e** present data from an entire ($8 \times 8 \times 3$) array. Shorted pixels are removed as non-representative data.

2 pixels off (only 64 pixels are active) to artificially introduce geometrical displacement for RGB pixels inherent for CFA images so that we could implement demosaicing algorithms for the resulting image. The simulated Bayer-pattern sensor returned a raw output image of a discouraging quality, highlighting the negative impact of demosaicing on the resolution of the final image (Fig. 4f). Furthermore, the image showed substantial colour imperfections. On the contrary, the image obtained in Foveon mode showed improved image quality without substantial colour artefacts, confirming the advantage of the stacked structure. When the sensor resolution was increased to $64 \times 64 \times 3$ pixels for the TFT imagers, the effect of demosaicing was not as obvious; however, the spatial resolution of the projected ring still decreased substantially (Extended Data Fig. 5j–m).

In summary, we fabricated perovskite photodetectors in a full-colour, monolithic vertically stacked architecture, offering a compelling alternative to widely used CFA and Foveon colour sensors (see the overall comparison in Table 1). In addition to the features described above, the stacked perovskite offers other advantages over CFA technology, including engineering the photoresponse exclusively in the visible region, eliminating the need for infrared filters. Furthermore, the integration of perovskite layers directly above the sensor readout electronics can substantially expand the light-sensitive area of the device, as was highlighted for CMOS sensors with integrated organic photodetectors[25]. An increase in resolution and light utilization may lead to an overall decrease in image sensor size, holding the potential to further miniaturize cameras in gadgets and reduce the complexity of camera lenses. Moreover, a realm of possibilities extends beyond contemporary consumer electronics and photography. With advancements in artificial intelligence and machine vision, the demand for sensors that can precisely identify objects by colour nuances indiscernible

**Table 1 | Comparative summary of Foveon, CFA (Bayer) and stacked-perovskite sensors**

|  | Silicon CFA | Foveon (stacked silicon) | Stacked perovskite |
| --- | --- | --- | --- |
| Colour reproduction | Fair | Bad | Good |
| Light utilization | Bad | Fair[a] | Good |
| Spatial resolution | Reduced | Original | Original |
| Demosaicing related artefacts | Strong | Minor | Minor |
| Detectivity | Good | Good | Under development[b] |
| Technological complexity | Relatively easy | Relatively complicated | Under development[b] |

The stacked-perovskite photodetectors outperform the conventional CFA image sensors in all important characteristics, except for the technological complexity of fabrication, which may be challenging to estimate yet. Although the Foveon and stacked-perovskite arrays have a rather similar set of characteristics owing to matching geometrical structure, the stacked-perovskite detectors have much better colour reproduction and more distinct light utilization. These advantages confirm the potential of monolithically stacked-perovskite detectors to overcome the fundamental limitations of the existing CMOS silicon sensors and ultimately to achieve a breakthrough in image quality and sensor performance.

[a]Good absorption, but photons are often absorbed in a wrong colour pixel.

[b]Difficult to compare at a research prototype stage.

to the human eye is growing. Such specifications might be uniquely addressable by perovskites owing to their sharp and finely tunable absorption edges.

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

# Methods

## Chemicals
All chemicals were used as supplied, without further purification.

**Perovskite precursors.** Methylammonium bromide (99.99%) was purchased from Great Cell Solar, caesium chloride (99.9999%) and caesium iodide (99.9999%) were purchased from Chemcraft, lead bromide (>98%) was purchased from Sigma) and lead iodide (99%) was purchased from Thermo Scientific.

**Sputtering targets.** Targets were obtained from Angstrom Engineering: silicon dioxide ($SiO_2$) sputtering target Ø3″, purity 99.995%, thickness 1/8″, bonded to 1/8″ OFHC backing plate. Tin oxide ($SnO_2$) 3″ diameter sputtering target, purity 99.9%, thickness 1/8″, bonded to 1/8″ oxygen-free high conductivity copper backing plate. ITO (90:10 wt% $In_2O_3$:$SnO_2$) hollow cathode sputtering target diameter 147 mm × height 52 mm, purity 99.99% indium, bonded to OFHC backing plate.

**Solvents.** Anhydrous (AcroSeal) ethanol and chlorobenzene were purchased from Acros (99%) and used inside a nitrogen-filled glovebox.

**Charge-transport materials and electrodes.** The charge-transport materials and electrodes were as follows: molybdenum trioxide (99.99%, Alpha-Aesar), (2-(9H-carbazol-9-yl)ethyl)phosphonic acid (2PACz; >99%, Lumtec), ZnO ink N-10 flex (Avantama), 4,4′,4-tris(carbazol-9-yl) triphenylamine (TCTA; 99.5%, Lumtec), AZO (aluminium zinc oxide) ink N−21X flex (Avantama), NiO ink P-21X-flex (Avantama), (6,6)-phenyl $C_{61}$ butyric acid methyl ester (PCBM; >99.5%, Lumtec) and $C_{60}$ (99.9% SES Research).

## Sensor fabrication
The substrates were subjected to a sequential cleaning process using Helmanex solution, water, acetone and isopropanol, after which they were treated with ultraviolet ozone for 10 min. Following this, thermal evaporation was employed to deposit 15 nm of molybdenum oxide, 30 nm of gold through a shadow mask and another 30 nm of molybdenum oxide. The device assembly continued with spin-coating a hole transporting layer of 2PACz (1.2 mg ml$^{-1}$ in ethanol) at 3,000 rpm, which was followed by the co-evaporation of MABr (132 °C) and $PbI_2$ (0.48 Å s$^{-1}$) at 20 °C substrate temperature to create a MAPbBrI$_2$ perovskite layer. A 40-nm $C_{60}$ layer was then thermally evaporated at a rate of 0.5 Å s$^{-1}$. Zinc oxide nanocrystals (Avantama N10-Flex) were then uniformly applied over the surface by spin-coating at 2,000 rpm. ITO (180 nm) was sputtered using a shadow mask, followed by the deposition of 40 nm of $SnO_2$ and a 200-nm $SiO_2$ layer over the entire substrate area.

The fabrication of the second pixel started with a 180-nm ITO electrode layer sputtering through a shadow mask. Further depositions included the evaporation of 40 nm of $SnO_2$, 5 nm of $C_{60}$ and a 300-nm layer of the second perovskite CsPbBr$_2$I deposited at 20 °C substrate temperature through co-evaporation using CsI (0.61 Å s$^{-1}$) and PbBr$_2$ (0.6 Å s$^{-1}$) sources. Thermal evaporation was used to deposit 15 nm of TCTA and 30 nm of MoO$_3$. The deposition of layers continued with the spin-coating of the AZO nanocrystal solution at 2,000 rpm, followed by the sputtering of a 140-nm ITO layer using a shadow mask. The subsequent steps included sputtering 40 nm of $SnO_2$ and 200 nm of $SiO_2$.

The fabrication of the third pixel started with 120-nm ITO electrode layer sputtering. The substrates then underwent air plasma treatment for 30 s, after which a 2PACz solution (1.2 mg ml$^{-1}$ in ethanol) and a NiO nanocrystal solution (Avantama P-21) were sequentially spin-coated at 3,000 rpm. A 5-nm TCTA layer was then thermally evaporated, which preceded the co-evaporation of the CsPbBr$_2$Cl perovskite from CsCl (0.49 Å s$^{-1}$) and PbBr$_2$ (0.6 Å s$^{-1}$) sources at 20 °C substrate temperature.

The fabrication progressed with the coating of a 40-nm PCBM layer deposited from a 20 mg ml$^{-1}$ solution in chlorobenzene, at 2,000 rpm. After depositing 5 nm of $C_{60}$ via thermal evaporation, the device was spin-coated with ZnO nanocrystals at 2,000 rpm. The fabrication concluded with the sputtering of a 120-nm ITO layer using a shadow mask. The device active area for the stacked detector was 6 mm$^2$/9 mm$^2$/6 mm$^2$ for the red, green and blue channel pixels, as defined by shadow masks.

## Device characterization
**External quantum efficiency.** The EQE spectra were measured in the wavelength range of 300 nm to 700 nm, utilizing a QE system (Model QE-R from Enli Tech). The measurements were conducted under near-dark test conditions with a chopper frequency set at 210 Hz.

**Optical measurements.** Absorption spectra in the ultraviolet–visible range for perovskite thin films were obtained in transmission mode using a Jasco V670 spectrometer.

**Capacitance measurements.** Capacitances were calculated from impedance spectroscopy data obtained for each pixel of a stacked detector using an Autolab PGSTAT302N, with a 50 mV amplitude and 0 V d.c. offset in darkness.

**Current–voltage measurements.** The $I$–$V$ curves of devices were collected with Keysight B2920 SMU using a homemade photodetector testing set-up with an RGB LED (LC-10W RGB-C SERIES from LCFOCUS) as a light source (Supplementary Fig. 6). The light flux was calibrated by measuring photocurrents with an FDS1010 photodiode from Thorlabs with a known responsivity spectrum. All $I$–$V$ measurements were done in a nitrogen-filled glovebox. The $I$–$V$ sweeps were performed at 200 mV s$^{-1}$ rates, first under dark and then under illuminated (1 mW cm$^{-2}$) conditions.

**Transient photocurrent measurements.** Transient photocurrent measurements were performed using a Becker & Hickl BDL-SMN Series 473-nm pulsed diode laser with a repetition rate of 100 kHz and pulse width of 90 ps. The signal was amplified with a Femto HSA-X-I-2-40 wideband voltage amplifier with 160 ps rise–fall time and 40 dB fixed gain before sending it to a Tektronix MSO44 mixed-signal oscilloscope with 500 MHz bandwidth, 160 ps resolution and 6.25 GS s$^{-1}$ sampling rate. For these measurements, we prepared special samples with an area of 20 × 200 µm defined with lithographically patterned ITO (bottom) and metal electrode deposited with a shadow mask (top).

**Atomic force microscopy.** Atomic force microscopy imaging was performed using an NX-10 Park AFM with AC160TS tips in non-contact mode, at a 0.5-Hz scan rate over a 2 × 2 µm area, capturing 256 points per row.

**Focused-ion-beam cross-section scanning electron microscopy.** The scanning electron microscope image was acquired at 3-kV electron acceleration voltage and 0.1-nA current using FEI Helios Nanolab 660 FIB-SEM. Electron detection was facilitated through a through-the-lens detector in back-scattered electron mode, enhancing material contrast. Preceding image capture, the sample was tilted at an angle of 52° to the scanning electron microscope cone, directly facing the focused-ion-beam source, which was employed for sample cutting.

**Noise-equivalent power and specific detectivity.** The estimation of the noise-equivalent power was performed based on the methodology outlined in the literature[42]. In brief, the photodetector current spectral density dependence on frequency was measured using the spectral analyser (SR770), under modulated LED light at various incident light powers. This measurement was conducted using a spectral analyser (SR770), with the photodetector exposed to modulated LED light at

different levels of incident light power. The power of the incident light was measured using a Thorlabs photodiode (s130VC) paired with a PM100USB power meter. LEDs with red (626 nm), green (522 nm) and blue (461 nm) emission maxima were used for the red, green and blue photodetector layers, respectively. The electrical bandwidth was determined by estimating the full-width at half-maximum of the current spectral density peak, which was calculated by fitting the lowest intensity peak with a Lorentzian function. The peak magnitude of the current was equated to the photoresponse of the device. The signal-to-noise ratio was calculated as the ratio of the photoresponse to the noise floor. This ratio was then graphed as a function of the input light power, and the noise-equivalent power was identified as the input power that resulted in a signal-to-noise ratio of 1. Furthermore, the specific detectivity ($D^*$) was calculated using the formula

$$D^* = \frac{\sqrt{A \times BW}}{NEP}$$

where $A$ is the area of the respective sensor and BW is the bandwidth.

**Colour-fidelity measurements.** Colour fidelity was evaluated using a custom-built set-up on an optical table. All measurements were performed under ambient conditions with the samples exposed to air for no more than 2 hours. White-LED light was projected onto ColorChecker patches through lenses, focusing the beam on approximately half of the total patch area to minimize reflections from adjacent areas. The reflected light was then collected by the sensor under characterization. After a stabilization time of 5 s at a 0 V bias, we recorded the sensor's photocurrent response. Each pixel in the layered detector configuration was measured independently. For a more detailed description of these experiments, refer to Supplementary Note 3. To reconstruct the ColorChecker chart representation from the initial photocurrent readings, the values were combined in a matrix and normalized to the highest photocurrent for each colour channel. These normalized values were then transformed into brightness levels by scaling to a 255 maximum, and subsequently rounded to the nearest integer. We observed an enhancement in image quality when we applied a square-root transformation to these brightness values followed by renormalization. The ColorChecker image was rendered from the individual pixel data through a dedicated Python script. Adjustments for grey balance were performed on the final image by adjusting the ratios of R, G and B channels to 1:1:1 utilizing the colour patch 1, 4 (grey). Other RGB channels of the colour patches were adjusted using the coefficients obtained from the grey balancing procedure.

**Testing 8×8×3 cross-bar arrays.** The testing of the sensor array was carried out using a custom-made printed circuit readout board equipped with an array of electromechanical relays with a switching speed of 200 ms, controlled by an Arduino microcontroller. Linearity measurements for each pixel were conducted sequentially, first in darkness and then under narrow-band LED illumination, using 0.5 mA to 64 mA current density for red, green and blue LEDs, separately. For the estimation of colour crosstalk under narrow-band emission, we used interpolated values of photocurrent at photon flux ($3 \times 10^{14}$ photons s$^{-1}$ cm$^{-2}$) within the linear region for all 3 channels. To measure light reflected from the ColorChecker patches, identical procedures as for single site detectors were used for all pixel arrays. Photocurrent values were measured at 0 V bias. Shorted arrays lines were removed from statistics as outliers. In addition, photocurrents from the black (1–1) patch were subtracted from the rest of the patches for the CIELAB colour space calculations to correct for the optical imperfections of the measurement set-up.

To illustrate demosaicing principles, the following data manipulations were performed using custom Python scripts. One representative cross-bar array was used for imaging. Photocurrent values at 0 V from the $I$–$V$ characteristics were organized into 4 distinct $8 \times 8$ matrices for dark conditions and each coloured (RGB) illumination. Then, the responses for each channel were normalized against the total output of the channel, on a pixel-by-pixel basis. The following manipulations with the data were performed to render the images. The input images were loaded as $8 \times 8$ pixel, 3-channel BMP images and converted into a three-dimensional array of brightness values. Next, each array of normalized responses from the stacked pixels was multiplied by the brightness values array obtained from the initial image. For instance, the blue channel of the new image was computed as the sum of the responses of red, green and blue pixels to blue light as the channel of the input image. The resulting images were brightness corrected by normalizing them against the ratio of the maximum brightness of each channel from the initial image, divided by the maximum of the calculated response. The condition of white light was assumed to correspond to the concurrent exposure to all three RGB lights at the defined intensity. CFA simulation was conducted by selectively activating only one RGB layer per site within the $8 \times 8$ arrays, effectively 'deactivating' mismatching signals under a predefined $2 \times 2$ [[R, G][G, B]] mask applied repeatedly across the array, emulating real CFA pixel configuration. The conversion to CIELAB colour space involved normalizing the raw photocurrent data and then combining responses from a photosite to blue, green and red light into the three channels of an RGB image. For example, the response to blue light from an RGB photosite was calculated from normalized photocurrents and separated into three image channels.

**Imaging with TFT arrays.** The fabrication of TFT arrays was done utilizing commercially available $64 \times 64$ TFT substrates from Linkzill (SC-X-A064-2201). The red- and blue-light detector structures were fabricated identically to the stacked-detector structure 2 with minor changes. We did not use ALD (atomic layer deposition) SnO$_2$ owing to difficulties in removing this layer from the bonding pads; instead a simple C$_{60}$/bathocuproine electron transport layer was used. For the green detector, a p–i–n structure identical to the one used in the blue detector was utilized, as the TFT arrays were designed to measure only positive photocurrents. Arrays were encapsulated using SU-8 photoresist layer. The D50 illuminant light was used to illuminate the ColorChecker chart, with the light focused on the mechanically stacked three TFT arrays (R, G and B top array). For sensor calibration, black-and-white paper sheets were used to define the maximum and minimum current range. The data processing workflow is illustrated in Supplementary Fig. 9.

## Data availability

The authors declare that all relevant data supporting the findings of this study are available within the paper and its Supplementary Information, and the source data of the corresponding figures.

## Code availability

The code used for colour accuracy calculations and sensor image processing is available at: https://gitlab.ethz.ch/yakunins/vertically-stacked-monolithic-perovskite-colour-photodetectors.

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

**Acknowledgements** The work was financially supported by ETH Zürich through the ETH+ Project SynMatLab: Laboratory for Multiscale Materials Synthesis. We acknowledge A. Kanak for help with the synthesis of starting materials; E. Hartmann for help with colour-accuracy measurements; B. Turedi for X-ray diffraction measurements; and F. Krumeich for energy-dispersive X-ray spectroscopy analysis of perovskite films. This work was in part supported by the European Union through the Horizon 2020 research and innovation programme (ERC CoG Grant, grant agreement number 819740, project SCALE-HALO).

**Author contributions** S.Y., S.T. and M.V.K. developed the concept of this study. S.T., X.L. and D.P. performed the fabrication of samples and characterization. E.W. and L.L.A.F. performed the scanning electron microscopy measurements and the lithography for device fabrication. K.S. performed the noise-equivalent-power measurements and the specific detectivity evaluation. R.K. and F.F. assisted with the layer deposition and EQE measurements. G.J.M. designed and fabricated the reading-out PCB board for array testing. X.L. and G.M. performed the speed measurements. S.Y. developed the model for colour-accuracy evaluation. S.T., S.Y. and M.V.K. wrote the paper with the contribution of all co-authors. M.V.K., S.Y. and I.S. supervised the work. All authors discussed the results and commented on the paper.

**Funding** Open access funding provided by Swiss Federal Institute of Technology Zurich.

**Competing interests** S.T., D.P., E.W., G.J.M., I.S., S.Y. and M.V.K. are named as inventors on a patent application arising from this work. The other authors declare no competing interests.

**Additional information**
**Correspondence and requests for materials** should be addressed to Sergii Yakunin or Maksym V. Kovalenko.

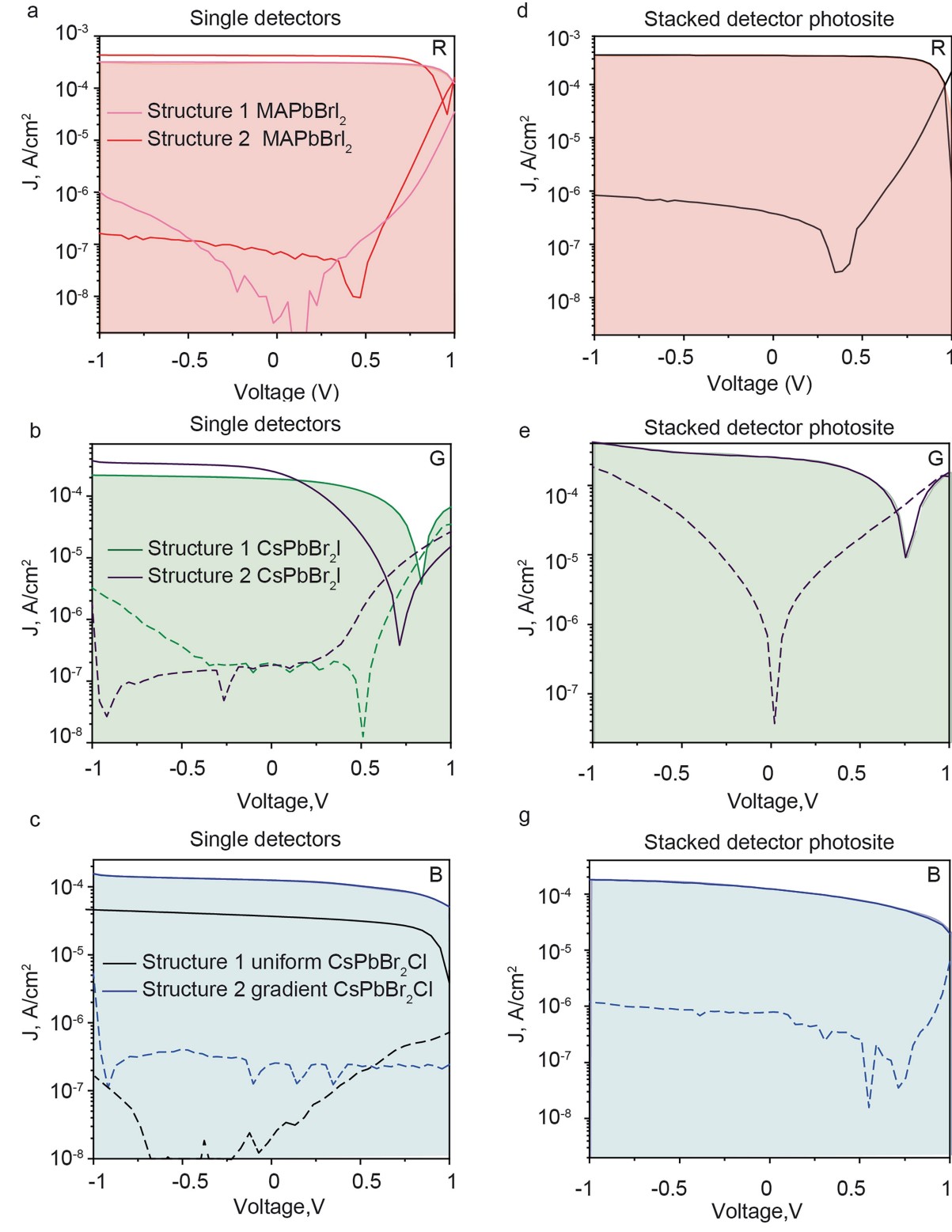

**Extended Data Fig. 1 | *J-V* characteristics of single-colour detectors with structures 1 and 2 and pixels of a stacked detector with structure 2.** The *J-V* characteristics of single-colour detectors for red (a), green (**b**), and blue (**c**) are presented. The structure of these single-colour detectors is identical to that of the detectors in the stacked configuration, except that the bottom electrode is made of transparent ITO (Kintec, 18 Ohm·sq⁻¹). The *J-V* characteristics of the stacked detector photosite with structure 2 are presented for red(d), green (e), and blue (g) pixels.

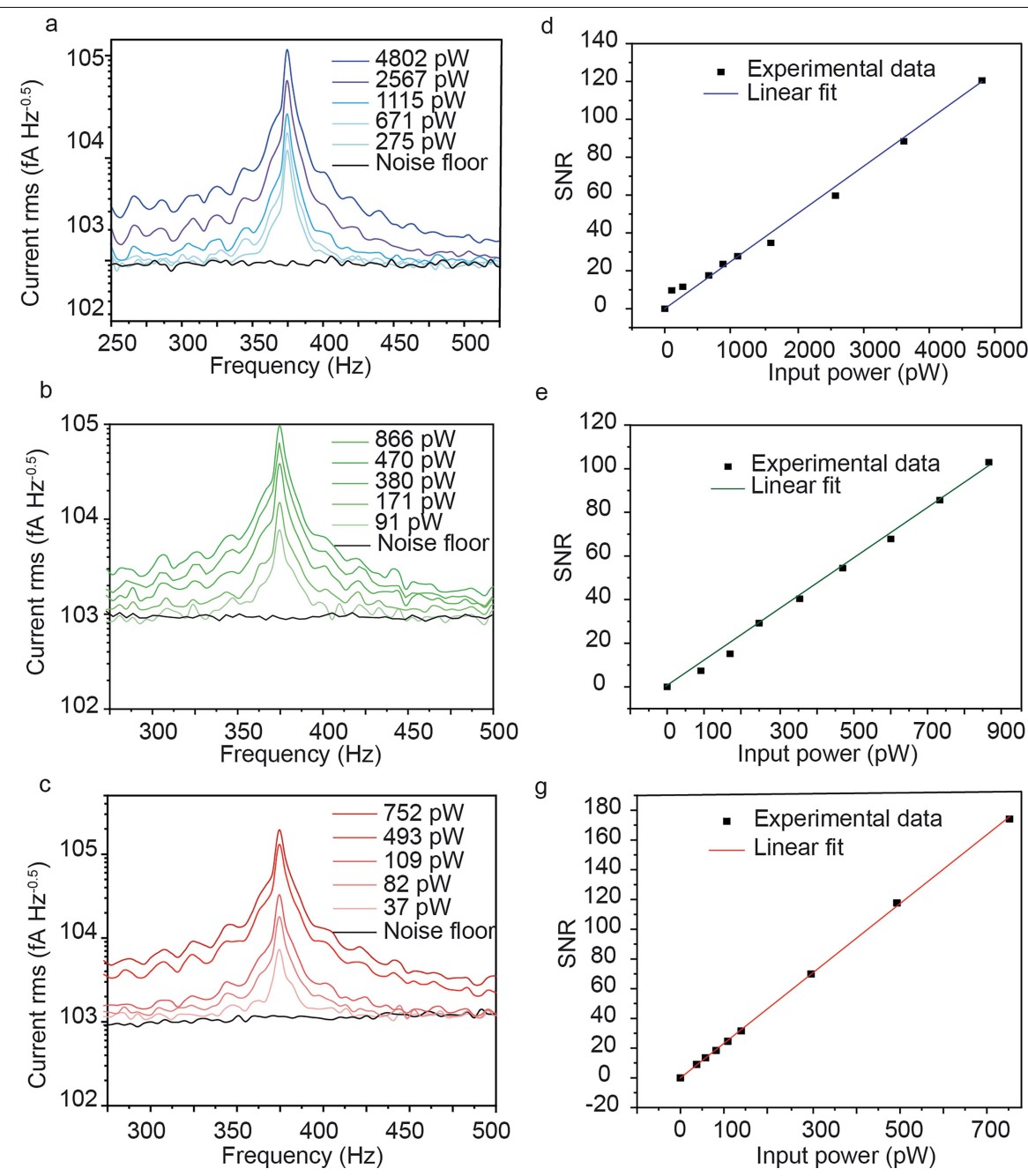

**Extended Data Fig. 2 | Noise spectral density and linearity of the stacked detector.** Red (**a, d**), Green (**b, e**), and Blue (**c, g**) pixels of the full-colour detector were measured under variable illumination levels and in dark conditions using corresponding RGB LEDs.

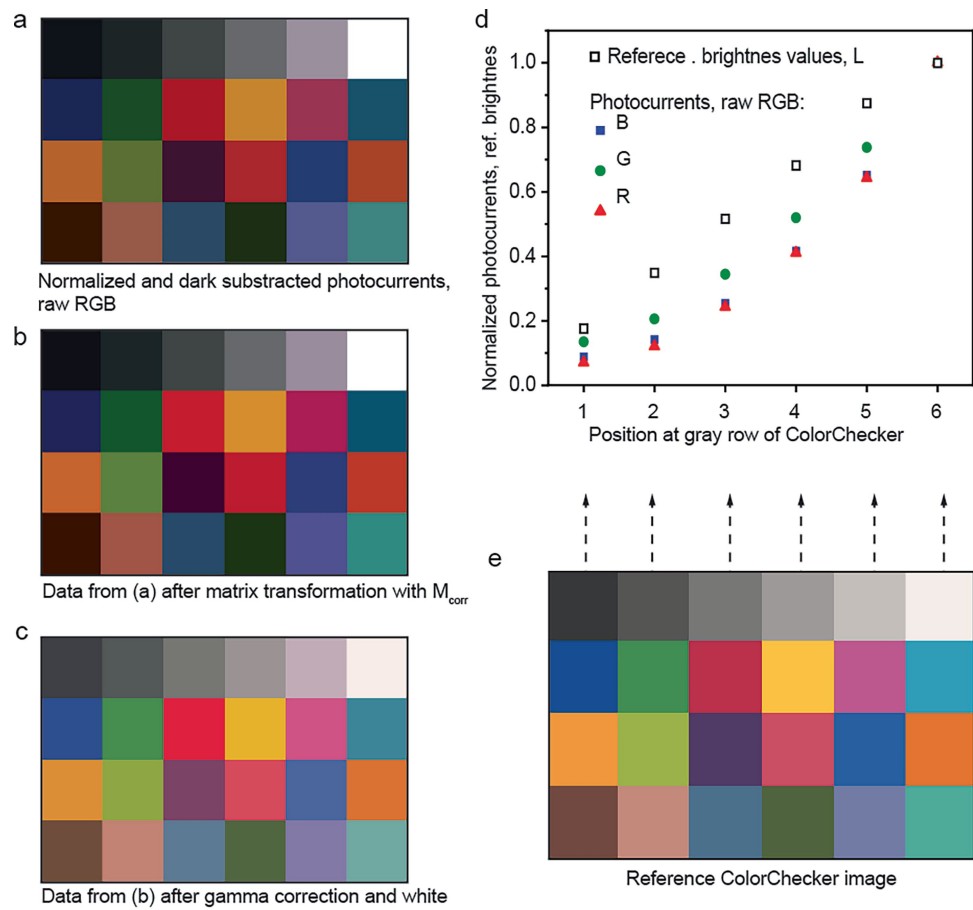

**Extended Data Fig. 3 | Intermediate steps in ColorChecker chart reconstruction from RGB photocurrent values.** We combined photocurrents from a stacked photosite into RGB channel brightness for reflected spectra from the ColorChecker patches. The photocurrents were normalized to the maximum current values for each channel. These values were subsequently rendered into an image representation of the ColorChecker. Notably, the resulting image exhibited a slight green tint (**a**), attributable to the larger area and higher sensitivity of the green pixel. To rectify the colour balance of the image, a grey balance procedure was employed (refer to Supplementary Note 3), leading to a reduction in the green tint (**b**). Furthermore, to enhance the brightness of the image, we applied a gamma correction function to de-linearize it (**c**). This approach accounts for the non-linearity of human visual response and display characteristics techniques[43]. **d**. The normalized reference luminance of colour patches, *vs*. brightness calculated from R, G, and B channels of the stacked detector.

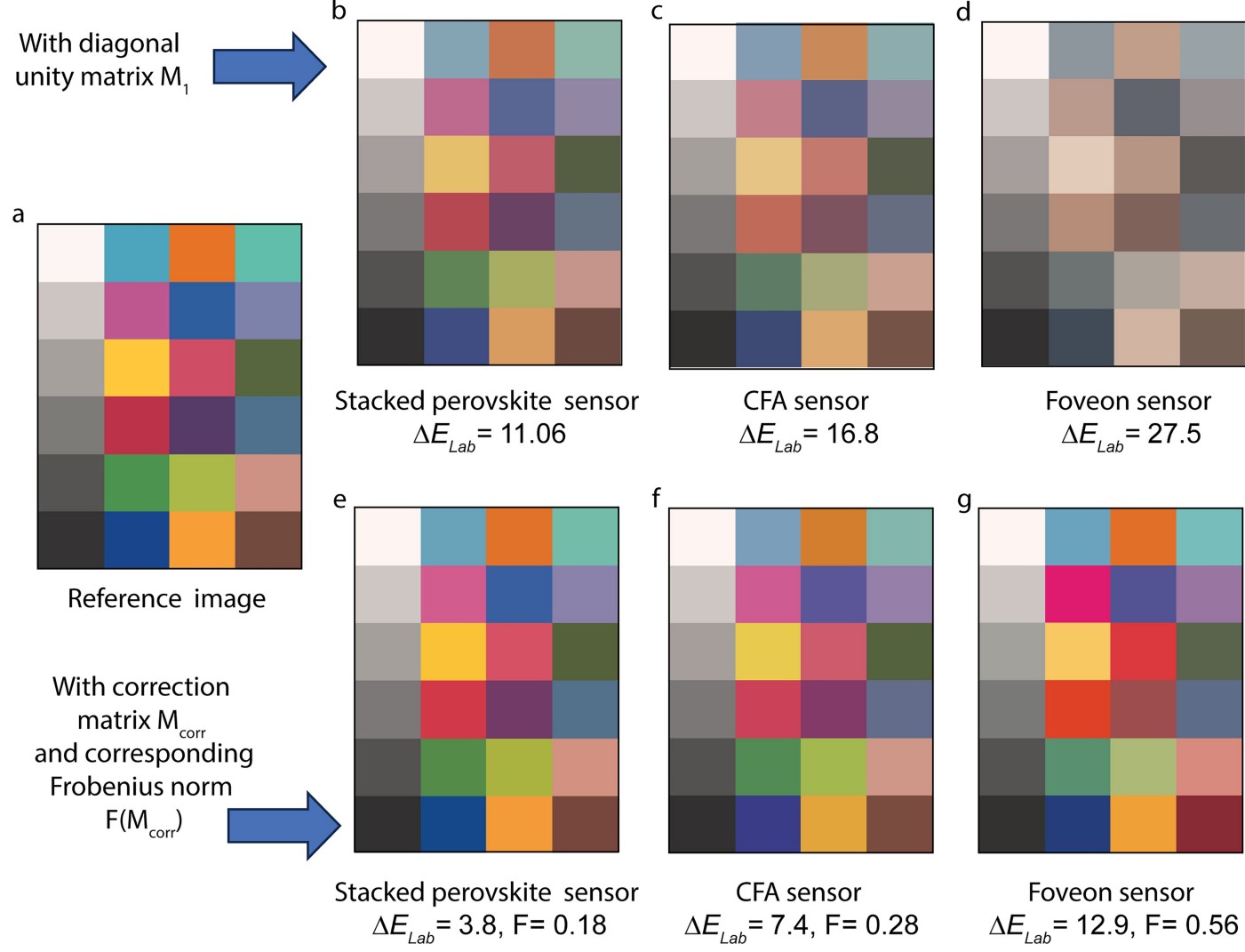

**With diagonal unity matrix $M_1$**

**a**

Reference image

**With correction matrix $M_{corr}$ and corresponding Frobenius norm $F(M_{corr})$**

**b**
Stacked perovskite sensor
$\Delta E_{Lab}$ = 11.06

**c**
CFA sensor
$\Delta E_{Lab}$ = 16.8

**d**
Foveon sensor
$\Delta E_{Lab}$ = 27.5

**e**
Stacked perovskite sensor
$\Delta E_{Lab}$ = 3.8, F= 0.18

**f**
CFA sensor
$\Delta E_{Lab}$ = 7.4, F= 0.28

**g**
Foveon sensor
$\Delta E_{Lab}$ = 12.9, F= 0.56

**Extended Data Fig. 4 | Colour accuracies calculated from spectral sensitivities of sensors following the Eqs. S1–S4 in Supplementary Note 3 and Supplementary Scheme 1. a**. An original image rendered from the reference values with indication of colour error $\Delta E_{Lab}$ (CIE1976 standard). **b-d**. The results obtained with diagonal unity matrix $M_1$. **b**. Stacked perovskite colour sensor presented in this work; **c**. CFA sensor Sony IMX249; **d**. Foveon™ sensor. **e-g**. Results obtained with correction matrices $M_{corr}$ (optimized for each sensor) and corresponding Frobenius norm.

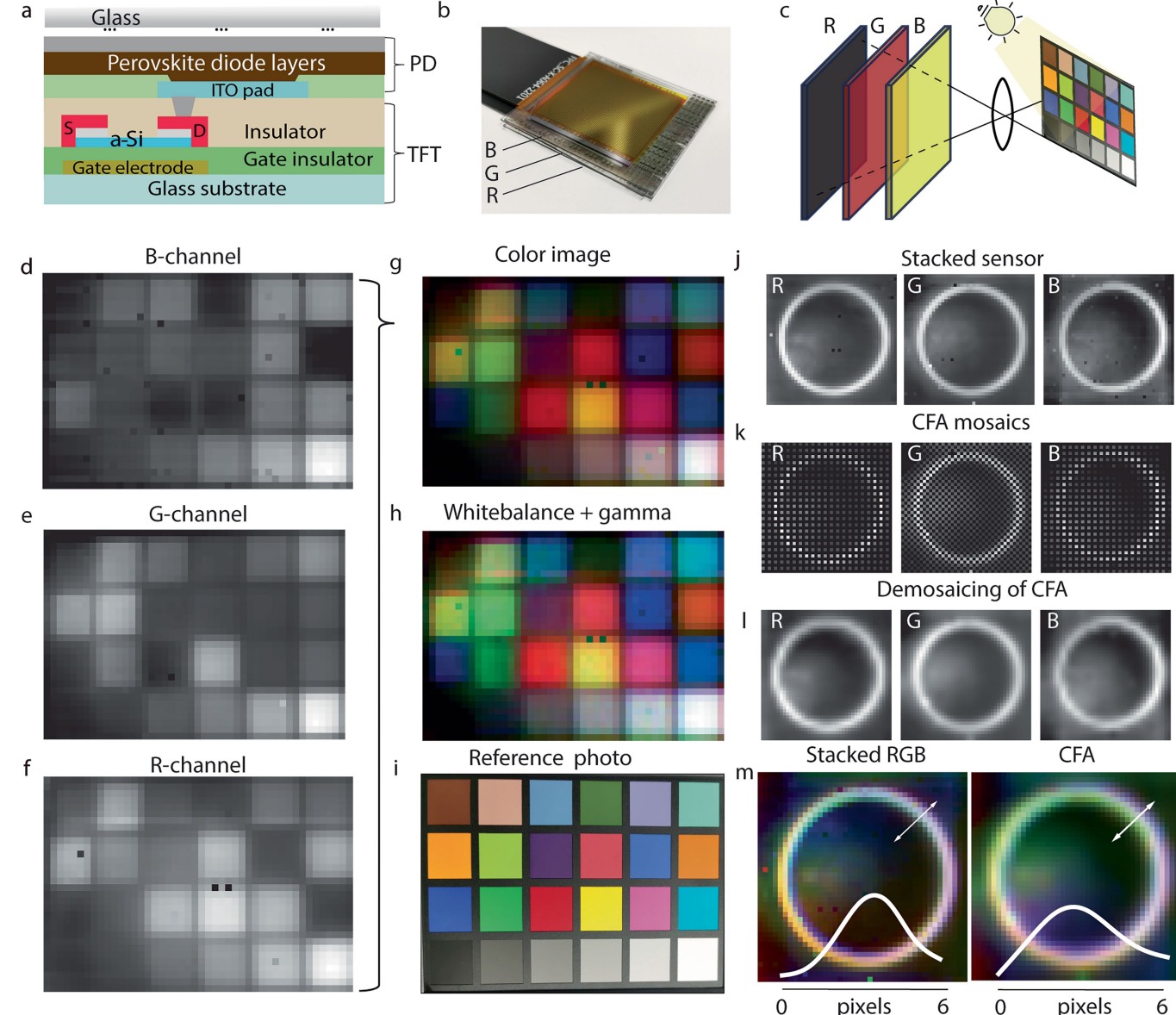

**Extended Data Fig. 5 | Photography of the ColorChecker chart and a white ring using stacked TFT array based camera.** Three semi-transparent arrays with a diode architecture similar to those used in the monolithically stacked architecture were utilised. **a**. Structure of a perovskite detector integrated with TFT pixel readout **b**. Photo of 3 mechanically stacked RGB perovskite TFT arrays **c**. Optical schema of photographic experiment. The D50 illuminant light reflected from a Macbeth ColorChecker chart was focused on the TFT array stack. **d-f**. B, G, R channel images of the ColorChecker obtained after flat-field correction (Supplementary Fig. 9). **g**. Combined three-channel colour image. **h**. The same image after white balance and brightness correction. i) reference photo of the ColorChecker chart. **j**. Images of a printed white ring on a black background obtained from the R, G, and B channels of the stacked TFT arrays. **k**. Imitation of CFA mosaics obtained by applying a numerical mask of Bayer pattern on (**j**). **l**. Result of demosaicing procedure applied on (**k**). **m**. Final colour image obtained from the stacked RGB and CFA sensors. Intensity profiles for both images are shown in brown (left) and purple (right).