## [Peer Review file · Nature]

Vertically stacked monolithic perovskite colour photodetectors

Corresponding Author: Professor Maksym Kovalenko

Version 1:

Reviewer comments:

Referee #1

(Remarks to the Author)

The authors present a novel color image sensor featuring perovskite-based vertically stacked photodiodes. With the color imaging sensor market exceeding 1 billion new devices annually, this technology has the potential to significantly enhance the quality of color images while providing a more sensitive device by improving quantum efficiency and color rendering accuracy. The authors fabricate and test a 2D array consisting of 8×8 vertically stacked pixels. Each pixel comprises three vertically stacked photodiodes, optimized to detect photons in the blue, green, and red spectra by utilizing perovskite materials with tailored energy bandgaps. The experimental results demonstrate enhanced quantum efficiency and improved color rendering performance.

There are several comments that the authors should address to enhance the quality and clarity of the presented data.

1. The statement below is partially correct and requires revision: 'These difficulties (from Foveon) arise from the silicon active layer's nearly uniform sensitivity across the visible spectrum, providing no inherent wavelength selectivity.'

While silicon's sensitivity does vary with wavelength due to its wavelength-dependent absorption coefficient, as previously mentioned by the authors, the issue lies in the nearly identical sensitivity of the three vertically stacked photodiodes. The spectral sensitivity arises primarily from the differing depths at which these photodiodes are placed within the silicon layer, rather than from intrinsic wavelength selectivity.

2. There has been noteworthy research on integrating perovskite with conventional photodetectors, as well as with Foveon's sensor for X-ray and UV+color imaging in the latter case. These studies are relevant to the work presented here and should be discussed in the manuscript.

Chen Q, Wu J, Ou X, Huang B, Almutlaq J, Zhumekenov AA, Guan X, Han S, Liang L, Yi Z, Li J. All-inorganic perovskite nanocrystal scintillators. *Nature*. 2018 Sep 6;561(7721):88-93.

Yakunin S, Sytnyk M, Kriegner D, Shrestha S, Richter M, Matt GJ, Azimi H, Brabec CJ, Stangl J, Kovalenko MV, Heiss W. Detection of X-ray photons by solution-processed lead halide perovskites. *Nature photonics*. 2015 Jul;9(7):444-9.

Chen C, Wang Z, Wu J, Deng Z, Zhang T, Zhu Z, Jin Y, Lew B, Srivastava I, Liang Z, Nie S. Bioinspired, vertically stacked, and perovskite nanocrystal-enhanced CMOS imaging sensors for resolving UV spectral signatures. *Science Advances*. 2023 Nov 3;9(44):eadk3860.

3. Current state of the art color image sensor use backside illuminated photodiodes providing 100% active area which is also covered by color filters. Hence the following statement should be moderated for accuracy: "Beyond their readily tunable bandgaps, perovskites offer advantages in coating 100% of the sensor's active area, which is significantly more challenging to achieve with traditional silicon-based technologies" The following paper provides excellent overview on color image sensors:

Oike Y. Evolution of image sensor architectures with stacked device technologies. *IEEE Transactions on Electron Devices*. 2021 Jul 26;69(6):2757-65.

4. Another advantage of the presented technology is the independent placement of the three photodiodes, unlike Foveon's technology. This flexibility allows the photodiodes to be positioned either next to each other for a compact configuration or at different depths to account for achromatic aberrations introduced by lenses. These aberrations can be particularly challenging to correct when imaging over a broad spectral range. It may be beneficial to highlight this advantage in the introduction.

5. It is important to distinguish the presented technology from Foveon's. Foveon's pixels utilize an active pixel design, integrating vertically stacked photodiodes with readout transistors, whereas the presented technology employs a passive pixel design. Active pixels offer advantages such as faster readout and lower noise. Including a discussion on how these benefits could be achieved with the proposed design would be valuable for the reader.

6. The data presented in Figure 3 is highly impressive, showcasing high-quality reconstruction. However, it would be valuable to include statistical analysis of the reconstruction accuracy for each color on the checkerboard. Given that the authors have fabricated a two-dimensional array, imaging each color pattern with the array and presenting the variation across all 64 pixels for the three photodiodes would provide important insights. Additionally, the authors should include imagery from each step of the processing pipeline. Presenting the raw data obtained from the image sensor prior to processing would allow readers to evaluate the initial quality of color rendering and better understand the impact of subsequent color processing. As one of the claims is simple processing of the raw data to generate accurate color image, showing intermediate color data will be critical.

7. The linearity of the three photodiodes, along with variations of the linearity across the 2D array, should be presented. Additionally, spatial variations in the array, such as fixed pattern noise, should be included for a more comprehensive analysis regarding the manufacturing variations.

8. Additional images collected with the sensor should be provided (main paper or supplementary). These images should demonstrate color replication under different illumination schemes.

Referee #2

(Remarks to the Author)

The manuscript reports on the design and fabrication of a stacked colour sensor using perovskites absorbers in wavelength-selective photodetectors. The fabrication process, based on vapour deposition of perovskites, and the characterisation of the photodetectors as well as colour accuracy and resolution of the full sensors are detailed.

Although the idea of using semiconductors with tailored band gaps as absorbers in stacked colour sensors has been proposed earlier (and cited by the authors), this work is a major step towards the realisation of commercially viable devices. The breakthrough is associated with the use of vapour deposition for the perovskites, which proves to be much more versatile compared to solution processing. While the performance of the individual photodetectors is slightly below champion devices reported in the literature, it is nevertheless very respectable.

The paper is well written and is generally backed by adequate measurements and data (see my comments below for minor improvements). Most of the remaining questions could be answered by consulting the supplemental information document. In general the paper is of high standard and conclusions are reasonably supported by the experiments.

Here's a list of suggestions to improve the manuscript:

- To reproduce this work it should include at least the deposition rates for the perovskites. I agree that other materials are relatively standard and do not need the same level of detail.
- It is well known that the composition of evaporated perovskites can substantially differ from the nominal composition obtained by combining the precursors. I would recommend an elemental analysis of the deposited films using either XRF or SEM-EDX.
- Calculation of EQE of the Foveon sensor from ref 11 (in Fig 1f) should be detailed in SI
- I think the 4-terminal structure mentioned in Suppl. Fig 2 is a 6-terminal one
- It would be nice to show a photograph of the actual 8x8x3 device
- The information contained in the CIELAB chart (Fig 4e) is relatively difficult to understand for a general audience. I'd suggest to give a bit more detail in the text to help the reader.
- Fig 2c-e. Caption does not explain what the dotted lines are (and why there are 2).
- Use of "Foveon-type" wording is a bit confusing as it applies to the Foveon X3 sensor as well as the stacked perovskites sensor. I'd suggest to use different wording like "stacked-type" for examples
- Fig 4f is not really specific to the new sensor developed in this work. It is a general feature of stacked sensors with respect to CFA sensors. It should probably go in supplemental info.
- Cesium chloride is mentioned twice in the list of precursors.
- Table 1 is very useful. While I agree that the detectivity should not be compared directly at this stage of development, it would be useful to comment on the potential to rival the other technologies on this aspect.
- What about combining technologies? For example use AlGaAs for red and deposit 2 layers of perovskites above?
- Why is structure 1 blue EQE much worse? This could be discussed in more details. I understand why the authors choose to show structure 2 in Fig 1i, but it would be more consistent to show structure 1 there as it is the one that is then discussed in the main text. Maybe they can be superimposed in Fig 1i?

Version 2:

Reviewer comments:

Referee #1

(Remarks to the Author)

The authors have address all of my concerns and comments. Thank you.

Referee #2

(Remarks to the Author)

The authors have taken into account the comments of both reviewers and addressed them adequately. Therefore I don't have any further comments.

Dear Editors, dear Referees,

We would like to thank the Referees for their careful reading and constructive feedback. We feel we were able to significantly improve the revised manuscript through additional experiments which involved fabrication and testing of TFT-integrated RGB stacked perovskite image sensors, measurements of stacked arrays linearity, perovskite layers characterization with XRD and EDX, recalculation of colour accuracy for arrays at various illumination conditions, re-testing cross-bar sensor arrays using reflected D-50 illuminant from ColorChecker patches. We rewrote the description of colour error calculation and added the results of the evaluation of various sensor types and illumination conditions. These results improved the accuracy of the paper, as well as its clarity and visual appeal. Furthermore, we note that this revised paper now provides the first demonstration of multipixel photography using vertically stacked active-pixel sensors since the inception of Foveon silicon sensors.

Below we detail the revisions made to the manuscript and our responses to the comments of all Referees. We thank the Referees for their in-depth and critical review of our manuscript as well as for their time, attention, and comments. While hoping for positive feedback, we will be grateful for further comments and suggestions.

Yours sincerely,

Maksym Kovalenko, on behalf of all authors.

Comments Editors and Referees are left in black, Authors' Answers are in blue; Actions are in green with revised manuscript text in yellow. All changes in the main or supplementary are commented on with an Action number. Replies are numbered (e.g. 1.x, 2.x, and for both Referees). To see the changes made to the manuscript, please see the comments and highlighted text in the revised manuscript. Those comments link to the numbered replies below.

Referees' comments:

Referee #1 (Remarks to the Author):

The authors present a novel color image sensor featuring perovskite-based vertically stacked photodiodes. With the color imaging sensor market exceeding 1 billion new devices annually, this technology has the potential to significantly enhance the quality of color images while providing a more sensitive device by improving quantum efficiency

and color rendering accuracy. The authors fabricate and test a 2D array consisting of 8×8 vertically stacked pixels. Each pixel comprises three vertically stacked photodiodes, optimised to detect photons in the blue, green, and red spectra by utilizing perovskite materials with tailored energy bandgaps. The experimental results demonstrate enhanced quantum efficiency and improved color rendering performance.

There are several comments that the authors should address to enhance the quality and clarity of the presented data.

Comment 1.1: The statement below is partially correct and requires revision: 'These difficulties (from Foveon) arise from the silicon active layer's nearly uniform sensitivity across the visible spectrum, providing no inherent wavelength selectivity, albeit its wavelength-dependent absorption coefficient.'

While silicon's sensitivity does vary with wavelength due to its wavelength-dependent absorption coefficient, as previously mentioned by the authors, the issue lies in the nearly identical sensitivity of the three vertically stacked photodiodes. The spectral sensitivity arises primarily from the differing depths at which these photodiodes are placed within the silicon layer, rather than from intrinsic wavelength selectivity.

Reply 1.1: We thank the Referee for pointing this out. Indeed, the Foveon sensor's poor spectral selectivity is primarily based on the wavelength-dependent absorption coefficient of Si. The manuscript text was altered to be more concise and clearer:

Action 1.1: A new text is added to the introduction section, lines 43-45 and lines 67-69.

Comment 1.2: There has been noteworthy research on integrating perovskite with conventional photodetectors, as well as with Foveon's sensor for X-ray and UV+color imaging in the latter case. These studies are relevant to the work presented here and should be discussed in the manuscript.

Chen Q, Wu J, Ou X, Huang B, Almutlaq J, Zhumeckenov AA, Guan X, Han S, Liang L, Yi Z, Li J. All-inorganic perovskite nanocrystal scintillators. *Nature*. 2018 Sep 6;561(7721):88-93.

Yakunin S, Sytnyk M, Kriegner D, Shrestha S, Richter M, Matt GJ, Azimi H, Brabec CJ, Stangl J, Kovalenko MV, Heiss W. Detection of X-ray photons by solution-processed lead halide perovskites. *Nature photonics*. 2015 Jul;9(7):444-9.

Chen C, Wang Z, Wu J, Deng Z, Zhang T, Zhu Z, Jin Y, Lew B, Srivastava I, Liang Z, Nie S. Bioinspired, vertically stacked, and perovskite nanocrystal-enhanced CMOS imaging

sensors for resolving UV spectral signatures. *Science Advances*. 2023 Nov 3;9(44):eadk3860.

Reply 1.2: Thank you for your suggestions. We included the first manuscript in our reference list in the introduction, where the key applications of perovskites are highlighted. However, as the 2nd and 3rd manuscripts are not directly connected to perovskite photodetectors, and furthermore the third manuscript actually characterises silicon photodiodes, and perovskite QDs are used as remitters, thus we decided to not include these in the main version of the manuscript

Action 1.2: Reference 1 is included in the reference list for the main text.

Comment 1.3. Current state of the art color image sensor use backside illuminated photodiodes providing 100% active area which is also covered by color filters. Hence the following statement should be moderated for accuracy: “Beyond their readily tunable bandgaps, perovskites offer advantages in coating 100% of the sensor's active area, which is significantly more challenging to achieve with traditional silicon-based technologie” The following paper provides excellent overview on color image sensors:

Oike Y. Evolution of image sensor architectures with stacked device technologies. *IEEE Transactions on Electron Devices*. 2021 Jul 26;69(6):2757-65.

Reply 1.3: Thank the Referee for pointing to this excellent review. Indeed, CMOS sensor technology has advanced significantly, and modern multichip stacked image sensors overcame many of the disadvantages of front-illuminated sensors. However, despite the availability of these stacked technologies, the ability to deposit the active layer directly onto the silicon readout integrated circuit (ROIC) would greatly reduce the technological complexity of chip stacking and bump bonding.

Action 1.3: A new text is added to the introduction section, lines 86-98.

Comment 1.4: Another advantage of the presented technology is the independent placement of the three photodiodes, unlike Foveon's technology. This flexibility allows the photodiodes to be positioned either next to each other for a compact configuration or at different depths to account for achromatic aberrations introduced by lenses. These aberrations can be particularly challenging to correct when imaging over a broad spectral range. It may be beneficial to highlight this advantage in the introduction.

Reply 1.4: We very much appreciate the Referee's comment, as we initially overlooked that this is an advantage of the stacked detector, as it requires a deeper knowledge of the optical objective technologies. In fact, depending on the type of objective, the chromatic aberrations can be a few μm (for compact smartphones or apochromatic

objectives) or a few tens of μm (for achromatic objectives). Varying the distance between the layers in the stack can compensate for the effects of chromatic aberration.

Action 1.4: A new text is added to the introduction section, lines 92-95.

Comment 1.5: It is important to distinguish the presented technology from Foveon's. Foveon's pixels utilise an active pixel design, integrating vertically stacked photodiodes with readout transistors, whereas the presented technology employs a passive pixel design. Active pixels offer advantages such as faster readout and lower noise. Including a discussion on how these benefits could be achieved with the proposed design would be valuable for the reader.

Reply 1.5: We very much agree with the Referee comment. In fact, in the present study, we present an experimental realization of monolithically stacked multilayer perovskite detector and detector arrays and illustrate their advantages over conventional one layer + optical filter design. The passive pixels are by no means a ready-to-use product, and in fact, the next step to integrate this design into a real-world camera with stacked pixels would be to use a combination of a transistor + photodiode to read out individual pixels and employ charge-integration for better sensitivity. It is worth noting that the presented RGB diode architectures are almost fully PVD and can easily be integrated on active pixel read-out arrays as we demonstrated further. However, a monolithic integration of ~ 30 layers structure with a TFT or CMOS ROIC is itself a very demanding task requiring the development of perovskite-specific lithography patterning methods, vertical interconnect, and also the design of a dedicated 3-channel image sensor, which itself is a very costly process that goes beyond the scope of this paper. Nonetheless, to give readers a grasp of how a real stacked camera will work, we constructed a multilayer mechanically stacked structure, employing three separate semi-transparent TFT image sensor substrates. This TFT 3-stacked sensor then was used to take a real photo of a Macbeth's ColorChecker chart.

Action 1.5: The detailed response is given further in response comments 1.7 and 1.8 and in Actions 1.7 and 1.8

Comment 1.6: The data presented in Figure 3 is highly impressive, showcasing high-quality reconstruction. However, it would be valuable to include a statistical analysis of the reconstruction accuracy for each color on the checkerboard. Given that the authors have fabricated a two-dimensional array, imaging each color pattern with the array and presenting the variation across all 64 pixels for the three photodiodes would provide important insights. Additionally, the authors should include imagery from each step of the processing pipeline. Presenting the raw data obtained from the image sensor before processing would allow readers to evaluate the initial quality of color rendering and better understand the impact of subsequent color processing. As one of the claims is simple processing of the raw data to generate accurate color images, showing intermediate color data will be critical.

Reply 1.6: We appreciate the Referee's suggestion to include statistics on colour accuracy. Accordingly, we analysed the multipixel cross-bar array shown in Fig. 4e and have now included the corresponding statistical data in the main text. Specifically, we calculated the standard deviation from the mean values in the data arrays to assess the pixel-by-pixel reproducibility of the 2D cross-bar array of stacked RGB pixels, thereby gauging the precision of our fabrication process. Because we operate outside of clean-room conditions, our precision is objectively modest. Nonetheless, the absolute colour errors relative to the reference values in this particular instance are less critical, as we have already shown that errors ΔE_{Lab} of 3.8 (from EQE spectra) and 7.6 (from photocurrents data) can be achieved by applying the appropriate raw data matrix transformation.

Based on the referee's comment, we added CIE LAB colour values of the blue [2, 1], green [2, 2], and red [2, 3] patches of the Macbeth colour chart measured with cross-bar arrays using reflected light of a D50 illuminant (the same experiment that we previously performed for single stacked perovskite photosite, Fig.3a) on Fig4.e, and kept the cross-bar arrays responses to RGB LED as histograms. There is some variation in the pixel's colour accuracy, which we, however, attribute to non-perfect optical alignment, reflections from the test holder, and perhaps certain inhomogeneity and scattering within the active layers. Nevertheless, these are quite impressive for a proof-of-concept prototype obtained in a lab without clean-room conditions.

According to the Referee's request to improve the visual demonstration of the step-by-step work of the image processing algorithms we:

- Substantially rewrote Supplementary Note 3 and the corresponding part of the Main Text (Lines 165-219) including Supplementary Scheme 1 which explains the basic steps of colour accuracy evaluation procedure. We believe that the current form is more clear for a broad readership.
- We included intermediate results in visual form for single-photosite experiments of colour evaluation (Extended Data Fig. 3 for photocurrents, Extended Data Fig. 4 for EQE spectra also including data for CFA and Foveon sensors for comparison) as well as intermediate results from newly created RGB stacked TFT arrays (Extended Data Fig. 5 and Supplementary Fig. 9)

Action 1.6: Supplementary Table 6, Supplementary Table 7, Supplementary Fig. 5, Supplementary Fig. 9, Supplementary Scheme 1, Extended Data Fig. 3, Extended Data Fig. 4 and Extended Data Fig. 5 are added to the Supplementary Information and the Main Text accordingly.

Figure 4 b-d,e have been redone. Supplementary Note 3 and Main Text (Lines 174-230, 234-241) were rewritten.

Comment 1.7: The linearity of the three photodiodes, along with variations of the linearity across the 2D array, should be presented. Additionally, spatial variations in the array, such as fixed pattern noise, should be included for a more comprehensive analysis of the manufacturing variations.

Reply 1.7: We thank the Reviewer for this important comment. We observed that, indeed, at higher photon fluxes, cross-bar arrays exhibit reduced linearity, likely due to the inherent limitations of the cross-bar ITO structure. This structure introduces a long charge-collection path and substantial resistance at the distal ends of the charge-collecting stripes. Additionally, any misalignment in the shadow masks and the partial line-of-sight nature of the sputtering process can cause electrode blurring, further contributing to the observed spatial and linearity variations.

The linearity and spatial variations of the photocurrent in 64-pixel arrays are shown in Supplementary Fig. 8. Since each individual pixel intensity dependence was normalized by the value at maximal light flux the deviations at lower fluxes reflect pixel-to-pixel variations of linearity.

Action 1.7.

Supplementary Figure 8 has been added.

New text is added at lines 261-266, 274-279.

Comment 1.8: Additional images collected with the sensor should be provided (main paper or supplementary). These images should demonstrate color replication under different illumination schemes.

Reply 1.8: As also noted by Referee 2, the existing images do not provide significant information related to the properties of the sensor. Therefore, here, to give readers a better view of what the perovskite stacked technology can do we decided to fabricate three-colours perovskite semi-transparent TFT arrays and assemble them into 64x64x3 stacked image sensor. This concept allows to demonstrate abilities of the monolithically integrated arrays, albeit additional optical losses of reflection from air/glass interfaces. We made images of the entire Macbeth colour chart to show colour replication and also a white ring on a black background to demonstrate advantages over demosaicing approach. These are now illustrated in the Extended Data Fig. 5.

Additionally, we removed previous images obtained from the cross-bar arrays by individual illumination with RGB LEDs as less representative data. We believe that all together these measures should provide more comprehensive information to highlight the advantages of the stacked detector architecture.

We have also evaluated the variation of ΔLab under different illumination schemes *via a* model with illuminant source, absorber, and EQE spectral dependencies. This way we can quantitatively compare CFA, Foveon, and Perovskite stack detectors for a broad range of illumination cases.

Action 1.8: The following figures have been added to the main text:

Extended Data Fig. 5, Supplementary Fig. 9. Text changes are added at lines 258, 261-266, 269 onwards, 303.

Referee #2 (Remarks to the Author):

The manuscript reports on the design and fabrication of a stacked colour sensor using perovskites absorbers in wavelength-selective photodetectors. The fabrication process, based on vapour deposition of perovskites, and the characterisation of the photodetectors as well as colour accuracy and resolution of the full sensors are detailed.

Although the idea of using semiconductors with tailored band gaps as absorbers in stacked colour sensors has been proposed earlier (and cited by the authors), this work is a major step towards the realisation of commercially viable devices. The breakthrough is associated with the use of vapour deposition for the perovskites, which proves to be much more versatile compared to solution processing. While the performance of the individual photodetectors is slightly below champion devices reported in the literature, it is nevertheless very respectable.

The paper is well written and is generally backed by adequate measurements and data (see my comments below for minor improvements). Most of the remaining questions could be answered by consulting the supplemental information document. In general, the paper is of high standard and conclusions are reasonably supported by the experiments.

Reply: We thank Referee #2 for the very positive assessment of our manuscript and for their critical comments. Now, through additional experiments, we are confident that we can further convince the Referee of the validity of our interpretations and suggestions.

Here's a list of suggestions to improve the manuscript:

Comment 2.1: To reproduce this work it should include at least the deposition rates for the perovskites. I agree that other materials are relatively standard and do not need the same level of detail.

Reply 2.1: We thank the Referee for noting this. We put the deposition rates into the manuscript

Action 2.1: Deposition rates were included in the manuscript, lines 359, 368, 377.

Comment 2.2: It is well known that the composition of evaporated perovskites can substantially differ from the nominal composition obtained by combining the precursors. I

would recommend an elemental analysis of the deposited films using either XRF or SEM-EDX.

Reply 2.2: We thank the Referee for pointing this out. Following the Referee's advice, we performed SEM-EDX analysis of our nominal compositions MAPbI₂Br, CsPbBr₂I and CsPbBr₂Cl (gradient and uniform depositions) films. The films were deposited on substrates identical to the ones used in the devices

From the EDX results it looks like the film composition did not totally resemble the nominal composition, possibly due to a halide exchange and off-stoichiometry conditions happening in the chamber. The resulting compositions of the films, according to the EDX, were MAPbBr_{0.8}I_{2.2}, Cs_{0.99}Pb_{1.1}Br_{1.9}I_{1.1}, Cs_{0.85}Pb_{1.1}Br_{1.5}Cl_{1.5} for uniformly co-evaporated films, and CsPb_{0.825}Br_{1.6}I_{1.4} for the films with a vertical gradient. Interestingly, depositing nominally equal composition for perovskite. Interestingly, for all of our chloride-bromide perovskite films, the deposition growth condition was in slightly Cs-rich conditions (0.49 Å/s CsCl vs 0.6 Å/s PbBr₂, while 0.46:0.6 Å/s rate ratio is necessary for nominally stoichiometric films), however, the resulting films resulted in a Cs-poor perovskite composition. The gradient deposition first included depositing a Cs-rich region that seemingly facilitated Cs-rich growth, which may result in better uniformity and crystallinity of the films. Furthermore, we also performed XRD scans of the CsPbBr₂Cl films, which found that the preferred orientation of the film changed. More on this can be found in the response to the comment 2.13

Action 2.2: Added Supplementary Table 3. In the Main Text added lines (121-123).

Comment 2.3: Calculation of EQE of the Foveon sensor from ref 11 (in Fig 1f) should be detailed in SI

Reply 2.3: We thank the Referee for finding this mistake in the text. In fact, the spectral responsivity was taken from ref 3, that directly characterised the EQE of the Foveon sensor, not the reference 11.

Action 2.3: Data were taken from Ref. 3 replaced the previous reference 11 at line 72.

Comment 2.4: I think the 4-terminal structure mentioned in Suppl. Fig 2 is a 6-terminal one

Reply 2.4: Even though we use separate electrode depositions for contacting multiple layers, the top red and bottom green, and top green and bottom blue electrode pairs are connected outside of the pixel active area, which results in 4 terminals in total. Such architecture is used as we found that devices without dielectric spacers showed increased leakage, possibly due to edge defects on the pixels

Comment 2.5: It would be nice to show a photograph of the actual 8x8x3 device

Reply 2.5: We thank the Referee for providing this suggestion. Photographs of TFT arrays and cross-bar arrays are included as a Supplementary Figure 7 now

Action 2.5: Supplementary Figure 7 is added.

Comment 2.6 The information contained in the CIELAB chart (Fig 4e) is relatively difficult to understand for a general audience. I'd suggest to give a bit more detail in the text to help the reader.

Reply 2.6: Due to the Referee comment we decided to add more explanation text on the reasoning and nature of the CIELAB space

Action 2.6: New text is added at line 269.

Comment 2.7 Fig 2c-e. Caption does not explain what the dotted lines are (and why there are 2).

Reply 2.7: We thank the Referee for careful reading and noticing the missing description.

Action 2.7: Fig. 2 caption was changed according to the Referee's comment.

Comment 2.8: Use of "Foveon-type" wording is a bit confusing as it applies to the Foveon X3 sensor as well as the stacked perovskites sensor. I'd suggest to use different wording like "stacked-type" for examples

Reply 2.8: As the Referee suggests, we replaced the "Foveon-type" to stacked-type

Comment 2.9: Fig 4f is not really specific to the new sensor developed in this work. It is a general feature of stacked sensors with respect to CFA sensors. It should probably go in supplemental info.

Reply 2.9: We thank the Referee for providing this suggestion. This is a correct statement, however, the stacked sensors, in general, are quite underused, so we think it would be beneficial for a general reader to know this advantage as well, even though it is not unique to only perovskite sensors.

Comment 2.10: Cesium chloride is mentioned twice in the list of precursors.

Reply 2.10: We thank the Referee for finding this mistake in the methods section. We removed the second mention

Action 2.10: The second mention of CsCl is removed from the precursor list.

Comment 2.11: Table 1 is very useful. While I agree that the detectivity should not be compared directly at this stage of development, it would be useful to comment on the potential to rival the other technologies on this aspect.

Reply 2.11: Thank the Referee for pointing this out. We do have an additional supplementary table 4 (Supplementary Table 3 in the old version of the manuscript) discussing potential rival technologies. The table is linked to the main text.

The detectivity observed in our data for the complex stacked architecture lags only slightly behind the best values obtained for solution-processed single-diode perovskite detectors (Supplementary Table 4).

Comment 2.12: What about combining technologies? For example use AlGaAs for red and deposit 2 layers of perovskites above?

Reply 2.12: We agree with the Referee that there is no fundamental reason to not combine different materials for stacking the active layers. Since the ultimate goal will be making CMOS or CCD-integrated stacked detector chips, we believe that the direct growth of III-V material on the Si ROIC will be prohibitively challenging. If that, however is demonstrated, the most desired path forward then would be to add two more III-V layers. However, we rather believe that the integration of perovskite + organic layers may be more feasible and practical. The advantages of such a hybrid technology may include having an upper organic blue-sensitive layer acting also as a moisture-repellent coating. Furthermore, integrating NIR-sensitive organics may open more exciting possibilities for NIR/VIS multispectral detectors without using oxygen-sensitive tin perovskites.

Action 2.12: an extra sentence is added to the supplementary info, line 153.

Comment 2.13: Why is structure 1 blue EQE much worse? This could be discussed in more details. I understand why the authors choose to show structure 2 in Fig 1i, but it would be more consistent to show structure 1 there as it is the one that is then discussed in the main text. Maybe they can be superimposed in Fig 1i?

Reply 2.13: We thank the Referee for this comment. The main difference between structures 1 and 2 lies in the composition of the CsPbBr₂Cl perovskite layer. We found that employing a gradient deposition process for CsPbBr₂Cl significantly improves the blue-light detector's external quantum efficiency (EQE)—likely due to changes in perovskite composition and crystallinity. We attribute this improvement to the formation of a bilayer comprising Cs-rich and Cs-poor regions.

Evidence for this bilayer structure comes from both SEM cross-sectional images and X-ray diffraction (XRD) data. In the SEM secondary-electron cross-sections, two distinct layers are visible; these correlate with broader XRD peaks observed for the gradient-deposited film (Fig. 4c). In contrast, the uniformly evaporated film shows a stronger (110)

texture (Fig. 4a). However, the gradient-deposited film exhibits a more random orientation, as indicated by the presence of both (100) and (110) diffraction peaks.

Overall, these findings suggest that the bilayer structure can enhance charge transport in thicker films (up to ~400–413 nm), thereby improving charge extraction in devices. Similar perovskite architectures have been reported to exhibit preferential n-type (A-site cation-poor) and p-type (A-site-rich) doping, which can create an “electron funnel” effect and further boost device performance.¹

Action 2.13: Structure 1 EQE was added to Fig. 1i, and figure captions were changed accordingly.

Supplementary Fig. 7 was added.

New Text is added to Supplementary Information, line 128-137.

References

- 1 Cui, P. *et al.* Planar p–n homojunction perovskite solar cells with efficiency exceeding 21.3%. *Nat. Energy* **4**, 150-159 (2019).